# TTLL12 is required for primary ciliary axoneme formation in polarized epithelial cells

Julia Ceglowski, Huxley K Hoffman, Andrew J Neumann, Katie J Hoff, Bailey L McCurdy,

Jeffrey K Moore [iD] & Rytis Prekeris [iD] ✉

## Abstract

**The primary cilium is a critical sensory organelle that is built of axonemal microtubules ensheathed by a ciliary membrane. In polarized epithelial cells, primary cilia reside on the apical surface and must extend these microtubules directly into the extracellular space and remain a stable structure. However, the factors regulating cross-talk between ciliation and cell polarization, as well as axonemal microtubule growth and stabilization in polarized epithelia, are not fully understood. In this study, we find TTLL12, a previously uncharacterized member of the Tubulin Tyrosine Ligase-Like (TTLL) family, localizes to the base of primary cilia and is required for cilia formation in polarized renal epithelial cells. We also show that TTLL12 directly binds to the α/β-tubulin heterodimer in vitro and regulates microtubule dynamics, stability, and post-translational modifications (PTMs). While all other TTLLs catalyze the addition of glutamate or glycine to microtubule C-terminal tails, TTLL12 uniquely affects tubulin PTMs by promoting both microtubule lysine acetylation and arginine methylation. Together, this work identifies a novel microtubule regulator and provides insight into the requirements for apical extracellular axoneme formation.**

**Keywords** Epithelial Cells; Cilia; TTLL12; Tubulin; Methylation
**Subject Category** Cell Adhesion, Polarity & Cytoskeleton

## Introduction

Epithelial cells line the surfaces and cavities of the body and serve as a critical barrier between the external and internal environment. Importantly, these cells are polarized into distinct domains: the apical surface, which contacts the gaseous or aqueous external environment, and the basolateral surface, which contacts neighboring cells or an extracellular matrix (Martin-Belmonte and Mostov, 2008). One essential structure on the apical surface of most epithelial cells is the primary cilium, which serves as a sensor of the external environment and regulates cell signaling, growth, and tissue homeostasis (Anvarian et al, 2019). Defects in primary cilia

signaling or formation are associated with many multisystemic diseases called ciliopathies (Hildebrandt et al, 2011). Renal disease is a prominent feature of many ciliopathies, as defects in the primary cilia of the polarized epithelial cells that line the renal tubules lead to the development of kidney cysts and loss of kidney function (McConnachie et al, 2021). In addition, autosomal dominant polycystic kidney disease (ADPKD) is caused by loss-of-function mutations in several factors regulating ciliary signaling (Ma, 2021). Therefore, it is important to understand the mechanisms underlying cilia formation and signaling, especially in renal polarized epithelial cells.

The primary cilium is a microtubule-based structure that nucleates from the mother centriole, extends into the extracellular space, and is ensheathed by a distinct plasma membrane compartment (Reiter et al, 2012). Two pathways of primary ciliogenesis have previously been described (Sorokin, 1962; 1968). The first consists of the recruitment of regulatory factors and ciliary membrane to the nucleus-proximal centrosome, followed by growth of the ciliary axoneme. This structure eventually migrates toward the cell surface to fuse with the plasma membrane and this pathway is common in nonpolarized cell types. The second is common in polarized epithelia and first consists of centrosome migration to the apical cell surface, which coincides with a clearing in the apical cortical actin network. This is followed by the recruitment of regulatory factors and ciliary membrane, and finally growth of the ciliary axoneme. One of the key differences between these two pathways is whether axoneme extension begins in the interior of the cell or out into the extracellular space, after centrosome docking; therefore, they have been termed the intracellular and extracellular ciliogenesis pathways, respectively. While early steps of ciliogenesis have been well-described, there is less known about specific regulators of axoneme extension, especially in polarized epithelial cells, or if there are additional regulators required in the polarized/extracellular context where the growing axoneme is less protected.

While axonemal microtubules undergo polymerization and depolymerization, the cilium is a remarkably stable structure (Keeling et al, 2016). A defining feature of axonemal microtubules is that they contain several post-translational modifications (PTMs) that are thought to contribute to the stability and proper functioning of the cilium (Wloga et al, 2017). These include acetylation at α-tubulin Lysine 40, which is inside the microtubule lumen, catalyzed by αTAT1, and has been proposed to support microtubule integrity during

Department of Cell and Developmental Biology, University of Colorado Anschutz Medical Campus, Aurora, CO 80015, USA. ✉E-mail: rytis.prekeris@ucdenver.edu

bending (Portran et al, 2017; Shida et al, 2010). In addition, glutamylation and glycylation occur on the C-terminal tails of both α/β-tubulin, are exposed on the microtubule surface, and catalyzed by enzymes in the Tubulin Tyrosine Ligase-Like (TTLL) family (Rogowski et al, 2009; van Dijk et al, 2007). Subsequently, several TTLLs have been shown to localize to the centrosome, and TTLL3, TTLL5, TTLL6, TTLL8, and TTLL11 have all been found to regulate primary cilia formation and function (Gadadhar et al, 2017; He et al, 2018; Mathieu et al, 2021; van Dijk et al, 2007).

Previous work in our lab has characterized the small GTPase, Rab19, as necessary for ciliation in both intracellular and extracellular ciliogenesis. We found that in polarized renal epithelial cells Rab19 is required for both the apical cortical actin clearing mediated through HOPS complex binding and initial ciliary membrane recruitment with binding partner TBC1D4 (Jewett et al, 2021). This surprisingly multifaceted role of Rab19 led us to ask if there were any novel Rab19 interacting proteins that may also regulate axoneme extension. Through our previous mass spectrometry screen for Rab19-binding partners, we identified an interesting Rab19-binding candidate, Tubulin Tyrosine Ligase-Like 12 (TTLL12). TTLL12 is an under-explored member of the TTLL family, as it does not have a reported enzymatic function, and the role of TTLL12 in regulating microtubules remains unknown. Therefore, in this study, we sought to define the function of TTLL12 in cilia formation in polarized epithelial cells.

Here, we find that TTLL12 directly binds the α/β-tubulin heterodimer via its TTL domain. Our data also suggest that the PTMs TTLL12 regulates are tubulin acetylation and methylation, although whether this regulation is direct, remains to be fully determined. We also provide evidence that the interaction between TTLL12 and tubulin promotes microtubule stability and regulates microtubule dynamics. Importantly, we find a population of TTLL12 localizes to the centrosome in ciliated cells. Interestingly, while TTLL12 regulates ciliation timing and length in nonpolarized cells, it is ultimately not required for intracellular ciliogenesis. In contrast, TTLL12 is necessary for axoneme growth in polarized epithelia cells that utilize the extracellular ciliogenesis pathway. Overall, our study provides mechanistic insight into the function of TTLL12 and suggests that there are distinct requirements for axoneme extension in polarized epithelial cells.

## Results

### Identification of TTLL12 as a Rab19-binding partner

Ciliation in polarized epithelial cells consists of centrosome migration to the apical cell surface, followed by a cortical clearing of apical actin, recruitment of ciliary membrane, and growth of stable axonemal microtubules. Previously, we identified the small GTPase, Rab19, as a critical ciliogenesis factor that coordinates both cortical clearing and extension of ciliary axoneme through distinctive binding partners. Therefore, we asked if there were any novel Rab19-binding partners that are essential for the growth/stabilization of axonemal microtubules. We interrogated our previous Flag-Rab19 co-immunoprecipitation/mass spectrometry list (Jewett et al, 2021) for possible candidates and identified microtubule-related proteins that were enriched (>sixfold) over the IgG control (Fig. EV1A). We identified

numerous microtubule-interacting proteins, and TTLL12 was the most abundant polypeptide from this group (Fig. EV1A).

To confirm that TTLL12 binds to Rab19, we used an in vitro binding assay. Recombinant 6His-TTLL12 was purified from Sf9 cells and recombinant GST-Rab19 was purified from *E. coli* (Fig. EV1B). Because Rab19 is a small GTPase, we asked whether its nucleotide status would affect interaction with TTLL12. Therefore, Rab19 was locked with either the GDP (inactive Rab19) or GMP-PNP (non-hydrolyzable GTP analog; active Rab19) nucleotide, incubated with TTLL12, and precipitated with glutathione beads (Fig. EV1C). We observed a significant increase in TTLL12 binding (~fivefold) when Rab19 is present as compared to a GST-only control (Fig. EV1C,D). The interaction between TTLL12 and Rab19 did not appear to depend on nucleotide status since TTLL12 bound similarly to GDP and GMP-PNP GST-Rab19 (Fig. EV1C,D), which is consistent with our previous mass spectrometry results (Fig. EV1A). To further confirm that Rab19 binds to TTLL12 we immunoprecipitated either GFP-Rab19 or GFP-only from MDCK cells and blotted the precipitate for the presence of endogenous TTLL12. As shown in the Fig. EV1I, TTLL12 did co-precipitate with GFP-Rab19. Taken together, these results indicate that Rab19 directly binds TTLL12, but TTLL12 is not a canonical Rab19 effector protein.

TTLL12 is a particularly interesting Rab19-binding partner because it belongs to the TTLL family. The first tubulin-modifying enzyme discovered was Tubulin Tyrosine Ligase (TTL), which binds to the α/β-tubulin heterodimer and ligates tyrosine onto the tail of α-tubulin (Raybin and Flavin, 1975). Subsequent work has identified 13 other proteins that contain a TTL domain (Fig. 1A; Janke et al, 2005; van Dijk et al, 2007). These proteins bind to microtubules and add either glutamate or glycine to tubulin c-terminal tails via the TTL domain, although TTLL2 has been characterized as a glutamylase only by sequence similarity (Fig. 1A, denoted by **; Rogowski et al, 2009; van Dijk et al, 2007). Both glutamylation and glycylation are enriched on axonemal micro-tubules and are essential for ciliary function (Wloga et al, 2017). However, TTLL12 is the most unusual member of the TTLL family. Overexpressing TTLL12 in cells does not increase microtubule glutamylation or glycylation, and TTLL12 does not demonstrate glutamylase or glycylase activity in vitro (Rogowski et al, 2009; van Dijk et al, 2007). Thus, it has been proposed that TTLL12 is a pseudoenzyme and has an unclear relationship with microtubules (Brants et al, 2012). Therefore, in this study, we sought to further investigate TTLL12 and determine its potential role in ciliogenesis.

To gain insight into the function of TTLL12, we next performed a proteomics screen to identify TTLL12 binding partners. To that end, we generated a TTLL12 knockout (TTLL12 KO) cell line overexpressing GFP-TTLL12 in Madin-Darby canine kidney II (MDCK) cells (Fig. EV1E), and then GFP-TTLL12 was immuno-precipitated using an anti-GFP nanobody, which was followed by mass spectrometry to identify putative TTLL12-interacting proteins. We were able to identify Rab19 (Fig. 1B), further confirming that TTLL12 is a Rab19-binding protein in cells. Interestingly, the most abundant interaction was with TUBA1B, an isotype of α-tubulin (Fig. 1B). To further verify our mass spectrometry results, we performed a co-immunoprecipitation of GFP-TTLL12 followed by a western blot for α-tubulin (Fig. 1C). Indeed, TTLL12 binds to tubulin, and this includes both α- and β-tubulin (Fig. EV1F). Overall, we identify TTLL12 as a Rab19-binding partner and given

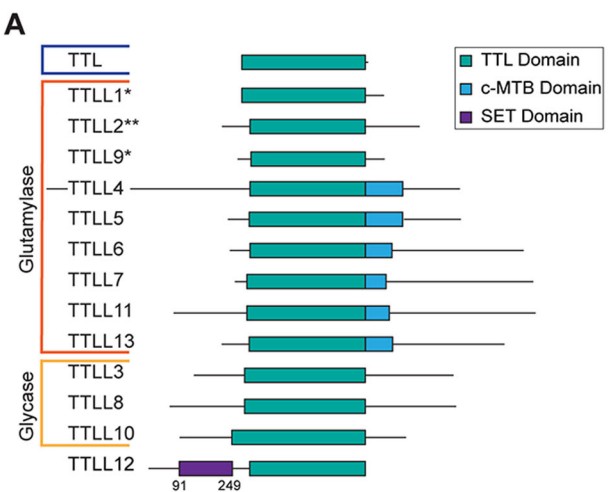

**A**

**B**

IP: GFP-TTLL12

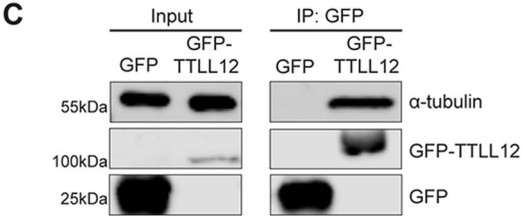

| Protein Identified | Spectral Counts | |
| --- | --- | --- |
| | GFP only | GFP-TTLL12 |
| TTLL12 | 22 | 294 |
| GFP | 1204 | 186 |
| Rab19 | 0 | 2 |
| TUBA1B | 0 | 367 |

**C**

**Figure 1. Identification of TTLL12 as a Rab19-binding partner.**

(A) Schematic of the TTLL family. Blue indicates TTL is a tyrosinase. *Indicates a glutamylase that functions in a complex. **Indicates a glutamylase that is classified only by sequence similarity. (B) Identification of TTLL12 binding partners. Spectral counts of candidate proteins identified through co-immunoprecipitation/mass spectrometry on GFP-TTLL12. The full list of proteins identified is in Dataset EV1. (C) TTLL12 binding to tubulin. Immunoprecipitation of GFP-TTLL12 followed by western blot for α-tubulin. Left column shows lysates (input) probed for GFP and α-tubulin. Right column shows immunoprecipitates probed for GFP and α-tubulin. Source data are available online for this figure.

the interaction with tubulin, a good candidate for regulating axonemal microtubules.

## TTLL12 directly binds to the α/β-tubulin heterodimer

Our data demonstrate that TTLL12 binds to tubulin (Fig. 1C). However, these experiments do not delineate whether this interaction

is with the microtubule polymer or the α/β-tubulin heterodimer. To test if TTLL12 directly interacts with the microtubule polymer, we used a microtubule co-sedimentation assay. Purified porcine brain tubulin was polymerized into microtubules and stabilized with Taxol. Microtubules were then incubated with recombinant purified 6His-TTLL12 and pelleted by centrifugation (Fig. 2A). TTLL12 did not significantly pellet with the microtubules (Fig. 2B), which suggests that TTLL12 does not directly bind microtubules. Brain tubulin is known to contain many PTMs. To eliminate the possibility that PTMs are preventing TTLL12 binding, we repeated the assay using yeast tubulin, which exhibits high conservation as compared to mammalian tubulin, but has no tubulin tail PTMs. In the absence of PTMs, TTLL12 still did not significantly pellet with yeast microtubules, further suggesting that TTLL12 does not interact with microtubule polymers (Fig. EV1G,H).

The lack of interaction between TTLL12 and the microtubule polymer may not be that surprising given the domain architecture of TTLL12. The microtubule-binding domain that has been identified in the glutamylases TTLL4, 5, 6, 7, 11, and 13 is located in the C-terminus after the TTL domain (Fig. 1A; Garnham et al, 2015). The TTLs without a C-terminal extension require the presence of other regulatory proteins to bind to microtubules (Fig. 1A, denoted by *; Janke et al, 2005; Kubo et al, 2014). TTLL12 also does not contain a C-terminal extension, so one possibility is that TTLL12 binds microtubules indirectly through an unidentified protein adaptor. However, another interesting possibility is that rather than directly interacting with microtubules, TTLL12 may bind to the α/β-tubulin heterodimer similarly to TTL. To test this, we performed a Ni-bead binding assay with recombinant purified 6His-TTLL12 and 0.18 μM untagged purified porcine brain tubulin at 4 °C to prevent microtubule polymerization (Fig. 2C). We find that when TTLL12 is present, there is a statistically significant increase (~threefold) in tubulin co-precipitation with Ni beads (Fig. 2D), which suggests that TTLL12 directly binds the tubulin heterodimer.

Next, we wanted to identify if the TTL domain is required for this interaction. To determine this, we purified GST-TTLL12 aa1-260, which lacks the TTL domain (Fig. 1A) and performed a GST pull-down assay with tubulin at 4 °C (Fig. 2E). GST-TTLL12 aa1–260 did not bind significantly more tubulin as compared to GST alone (Fig. 2F), which indicates that the TTL domain is required for TTLL12 to bind the tubulin heterodimer. Finally, we asked whether TTLL12 could bind Rab19 and tubulin together, or if these binding events were mutually exclusive. To test this, we performed a competitive Ni-bead binding experiment: 6His-TTLL12 and tubulin were incubated at 4 °C as before, but with increasing concentrations of GST-Rab19 (Fig. 2G). Interestingly, we find that Rab19 did not compete with tubulin for binding to 6His-TTLL12 (Fig. 2H), suggesting that, at least in vitro, TTLL12, Rab19, and tubulin can be part of the same protein complex. Overall, we conclude that TTLL12 is indeed a unique TTL that directly binds the tubulin heterodimer rather than the microtubule polymer. In addition, the TTL domain is required for tubulin binding, and TTLL12 can co-bind Rab19 and tubulin together.

## TTLL12 regulates microtubule stability and dynamics

Given that the enzymatic function of TTLL12 remains unknown, we next asked whether TTLL12 affects tubulin PTMs. We assessed tubulin PTMs by western blot in MDCK WT and TTLL12 KO cells.

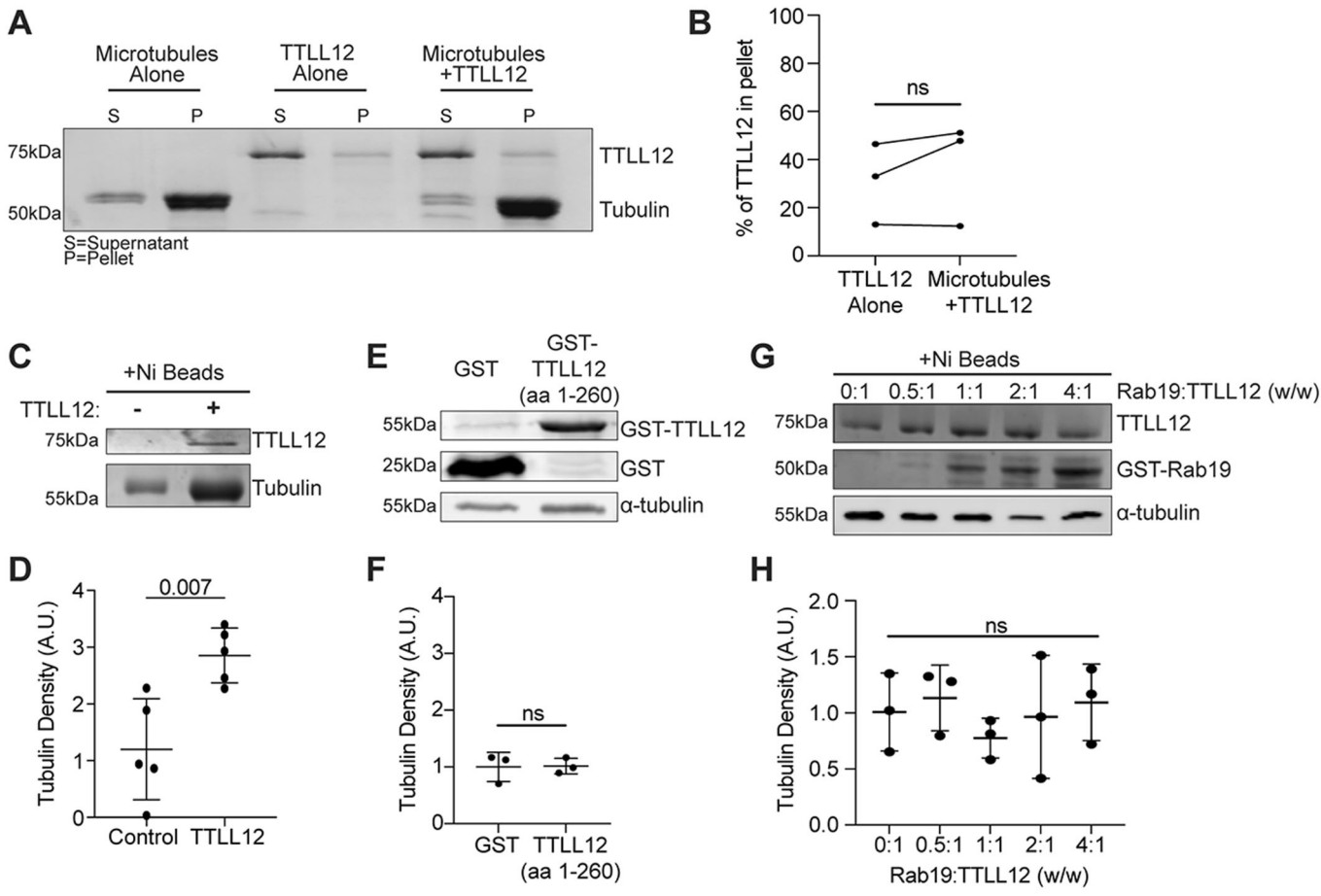

**Figure 2. TTLL12 directly binds to the α/β-tubulin heterodimer.**

(A) Microtubule and TTLL12 co-sedimentation assay. Taxol-stabilized porcine brain microtubules were mixed with recombinant 6His-TTLL12 and pelleted by centrifugation. The supernatant and pellet were separated and run on an SDS/PAGE gel followed by Coomassie blue staining. (B) Quantification of TTLL12 in the pellet with or without microtubules from (A). Graph shows individual values calculated from three independent experiments. Student's *t* test (two-tailed) was used for statistical analysis. (C) TTLL12 binding to the tubulin heterodimer. Binding assay with recombinant 6His-TTLL12 and porcine brain tubulin. TTLL12 and tubulin were mixed O/N at 4 °C to prevent microtubule polymerization, followed by the addition of nickel beads. Bead eluates were run on an SDS/PAGE gel followed by Coomassie blue staining. (D) Quantification of tubulin band intensity in (C). Graph shows mean ± SD derived from five independent experiments. Student's *t* test (two-tailed) was used for statistical analysis. (E) TTLL12 N-terminus binding to tubulin. Binding assay with recombinant GST-TTLL12 (aa1–260) and porcine brain tubulin. TTLL12 and tubulin were mixed O/N at 4 °C followed by the addition of glutathione beads and western blot probing for GST and α-tubulin. (F) Quantification of α-tubulin band intensity in (E). Graph shows mean ± SD derived from three independent experiments. Student's *t* test (two-tailed) was used for statistical analysis. (G) Competitive binding experiment between recombinant 6His-TTLL12, GST-Rab19, and untagged porcine brain tubulin. Increasing concentrations of GST-Rab19 were added to 6His-TTLL12 and tubulin followed by the addition of nickel beads. The presence of TTLL12 and Rab19 was confirmed by Coomassie and western blot was used to probe for α-tubulin. (H) Quantification of α-tubulin band intensity in (G). Graph shows mean ± SD derived from three independent experiments. Student's *t* test (two-tailed) was used for statistical analysis. Source data are available online for this figure.

We observed no difference in tyrosination or polyglutamylation (Fig. EV2A,B), but we did see a very moderate, but statistically significant decrease in acetylation in TTLL12 KO cells (Fig. 3A). Note that overexpression of GFP-tagged TTLL12 did not rescue this TTLL12 KO-induced loss of microtubule acetylation (Fig. 3A), although we did not determine whether this was due to addition of GFP-tag or is a consequence of TTLL12 indirectly affecting tubulin acetylation.

Acetylation typically is associated with stable microtubules since α-TAT1 (tubulin acetyltransferase) prefers binding microtubules while HDAC6 (tubulin de-acetylase) binds to tubulin α/β-heterodimer. Thus, we then asked if TTLL12 KO decreases

microtubule stability, thus, indirectly leading to decrease in acetylation. To test this, we incubated MDCK and MDCK TTLL12 KO cells at 4 °C for 30 min to destabilize microtubules, followed by fixing and staining with anti-α-tubulin antibodies (Fig. 3B). Since it is difficult to image and quantify microtubules in polarized columnar epithelial cells, we focused on mitotic MDCK cells to quantify microtubule stability in mitotic spindle. In addition, we observed that endogenous TTLL12 (in RPE1 cells) and TTLL12-GFP (in MDCK cells) are both present at the mitotic spindle centrosomes (Fig. EV3A,B), suggesting that TTLL12 may be involved in regulation of mitotic spindle microtubule stability. Consistent with this hypothesis, we observed a 50% decrease in

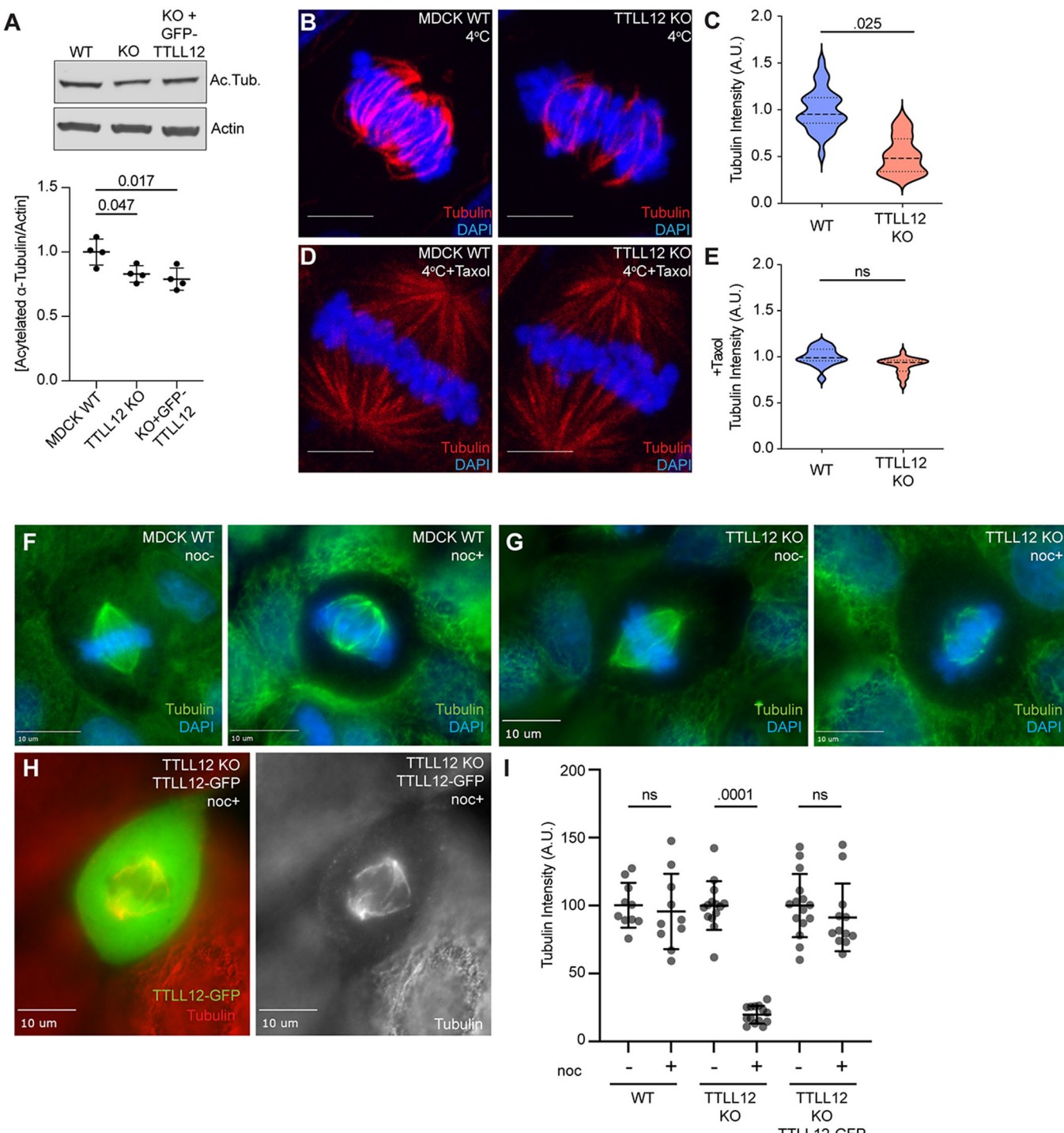

remaining microtubules in TTLL12 KO cells (Fig. 3C), which indicates that TTLL12 is required for microtubule stability. Importantly, the treatment of cells with the microtubule-stabilizing agent, taxol, rescued the effects of TTLL12 KO on cold-induced microtubule stability (Fig. 3D,E).

To further confirm that TTLL12 increases mitotic spindle microtubule stability, we next tested the effect of TTLL12 KO on sensitivity to microtubule-depolymerizing agent, nocodazole. To

that end we incubated MDCK cells for 10 min in the presence or absence of 500 nM of nocodazole. At these conditions, nocodazole has a mild effect on mitotic spindle microtubules in control MDCK cells (Fig. 3F,I). In contrast, this nocodazole treatment significantly decreased the number of mitotic spindle microtubules in MDCK TTLL12 KO cells (Fig. 3G,I). Importantly, overexpression of TTLL12-GFP partially rescued TTLL12 KO-induced sensitivity to nocodazole (Fig. 3H,I).

◄ **Figure 3. TTLL12 regulates microtubule stability and dynamics.**

(A) Representative western blot and quantification of acetylated α-tubulin in MDCK WT, TTLL12 KO, and TTLL12 KO + GFP-TTLL12 cells. $N = 4$. Graph shows mean ± SD. Student's $t$ test (two-tailed) was used for statistical analysis. (B) Microtubule cold-stability assay. Representative images of MDCK WT and TTLL12 KO cells placed at 4 °C for 30 min to depolymerize microtubules and stained for DNA and tubulin. Scale bar = 5 µm. (C) Quantification of tubulin intensity from (B). Data from three independent experiments ($N = 3$, $n = 30$ for WT, $n = 30$ for KO). Graph shows median and quartiles. Student's $t$ test (two-tailed) was used for statistical analysis. (D) Representative images of MDCK WT and TTLL12 KO cells treated with 1 µM Taxol before 4 °C for 30 min and stained for DNA and tubulin. Scale bar = 5 µm. (E) Quantification of tubulin intensity from Taxol-treated cells in (C). Data from three independent experiments ($N = 3$, $n = 19$ for WT, $n = 20$ for KO). Graph shows median and quartiles. Student's $t$ test (two-tailed) was used for statistical analysis. (F) Microtubule nocodazole sensitivity assay. Representative images of MDCK WT cells incubated in the presence or absence of 500 nM of nocodazole for 10 min and stained for DNA and tubulin. (G) Microtubule nocodazole sensitivity assay. Representative images of TTLL12 KO cells incubated in the presence or absence of 500 nM of nocodazole for 10 min and stained for DNA and tubulin. (H) Microtubule nocodazole sensitivity assay. Representative images of TTLL12 KO expressing TTLL12-GFP cells incubated in the presence of 500 nM of nocodazole for 10 min and stained for DNA and tubulin. (I) Quantification of tubulin intensity from (F–H). Each dot represents a single cell analyzed. Graph shows mean and standard deviation derived from three independent experiments. Student's $t$ test (two-tailed) was used for statistical analysis. Source data are available online for this figure.

Since TTLL12 regulates mitotic microtubule stability, we wondered whether TTLL12 may also affect interphase microtubule stability. To examine this, we first generated a TTLL12 KO cell line using CRISPR-Cas9 in human retinal pigmented epithelial (RPE1) cells (Fig. EV2C) because these cells are amenable to transfection and better suited for time-lapse microscopy than polarized columnar MDCK cells. Because TTLL12 KO decreased microtubule acetylation in MDCK cells, we first asked whether TTLL12 depletion has the same effect in RPE1 cells. We observed almost a complete ablation of microtubule acetylation in TTLL12 KO RPE1 cells (Fig. EV3C), but similarly to MDCK cells, saw no difference in microtubule tyrosination or polyglutamylation (Fig. EV2D,E). Next, we transiently transfected cells with GFP-MACF18, which binds and tracks the growing plus-ends of microtubules, and measured the number (to assess number of growing microtubules) and speed (to assess rate of microtubule polymerization) of GFP-MACF18 comets (Fig. EV3D and Movies EV1 and EV2). The number of microtubule polymerization events over a 4-min timeframe was similar in WT vs TTLL12 KO cells (Fig. EV3E), suggesting TTLL12 KO does not affect the number of growing microtubules. In contrast, compared to WT, TTLL12 KO cells had a faster microtubule polymerization rate (Fig. EV3F), which suggests that TTLL12 may slow down microtubule assembly. In addition, we also measured the lifetime of the GFP-MACF18 comets, which indicates the duration of microtubule growth before a pause or depolymerization. More comets had a shorter lifetime in TTLL12 KO cells than in WT (Fig. EV3G). Overall, we find TTLL12 KO increases interphase microtubule assembly rates while decreasing the duration of interphase microtubule growth, likely leading to decreased microtubule stability. Interestingly, TTLL12 KO had a much milder effect on interphase microtubule stability as compared to mitotic spindle microtubules. Furthermore, while TTLL12 is present at mitotic spindle centrosomes, we did not detect TTLL12 at interphase centrosomes (except the basal body, see Fig. 4C). All these data suggest that TTLL12 may regulate microtubule stability in specialized microtubule structures, such as mitotic spindles or cilia.

## TTLL12 is not required for intracellular ciliogenesis but regulates ciliation timing and length

Since we identified TTLL12 as a candidate for regulating axonemal microtubules, we next asked whether and how TTLL12 might impact cilia formation and function. There are two primary ciliogenesis pathways that have previously been described, and one major difference between these pathways is whether axoneme growth begins in the interior of the cell before centrosome migration (intracellular) or out into the extracellular space after centrosome migration (extracellular). The intracellular ciliogenesis pathway is common in nonpolarized cell types, such as RPE1s. Given that acetylated tubulin is typically a marker of axonemal microtubules and there was a severe decrease of acetylated tubulin in the TTLL12 KO RPE1 cells, we first wanted to test if TTLL12 functions in the intracellular ciliogenesis pathway. To determine if TTLL12 is necessary for intracellular ciliogenesis, we serum-starved WT and TTLL12 KO RPE1 cells for 48 h and stained them for Arl13b, a ciliary membrane marker, and acetylated tubulin (Fig. 4A). Curiously, we observed that TTLL12 KO cells form primary cilia that contain Arl13b, but these cilia are devoid of acetylated tubulin (Fig. 4A). We then asked if the TTLL12 KO axonemes still acquire other canonical PTMs, such as glutamylation. Using the marker GT335, we saw the cilia in TTLL12 KO cells do have glutamylated tubulin, similar to WT cells (Fig. 4B), suggesting TTLL12 is not required for intracellular ciliogenesis but is necessary for axoneme microtubule acetylation.

Next, we considered three hypotheses for how TTLL12 may be mediating tubulin acetylation. First, we tested whether TTLL12 may be binding and directly inhibiting aTAT1. To that end, we immunoprecipitated TTLL12 from wild-type or TTLL12 KO RPE1 cells using an anti-TTLL12 antibody and immunoblotted for the presence of αTAT1. As shown in the Fig. EV2F, αTAT1 did not co-precipitate with TTLL12, thus, it is unlikely that TTLL12 directly binds αTAT1 and regulates its enzymatic activity. Since acetylation is associated with stable microtubules, and we found TTLL12 KO reduces microtubule stability, next we hypothesized that by increasing microtubule stability, we could rescue microtubule acetylation in TTLL12 KO cells. To test this, we treated WT and TTLL12 KO RPE1 cells with Taxol for 2 h and then imaged acetylated tubulin. We observed an increase in acetylated tubulin in WT, but not TTLL12 KO RPE1 cells (Fig. EV4A). Microtubules are acetylated by αTAT1 predominately in the microtubule polymer, while it is de-acetylated by HDAC6 at the α/β-tubulin heterodimer state. Since TTLL12 binds the α/β-tubulin heterodimer, we next hypothesized TTLL12 may negatively regulate HDAC6 by preventing its association with α/β-tubulin heterodimer. Consequently, TTLL12 depletion would lead to additional HDAC6 activity and decrease microtubule acetylation. To test this hypothesis, we treated RPE1 cells for 24 h with 1 µM of the pharmacological HDAC6

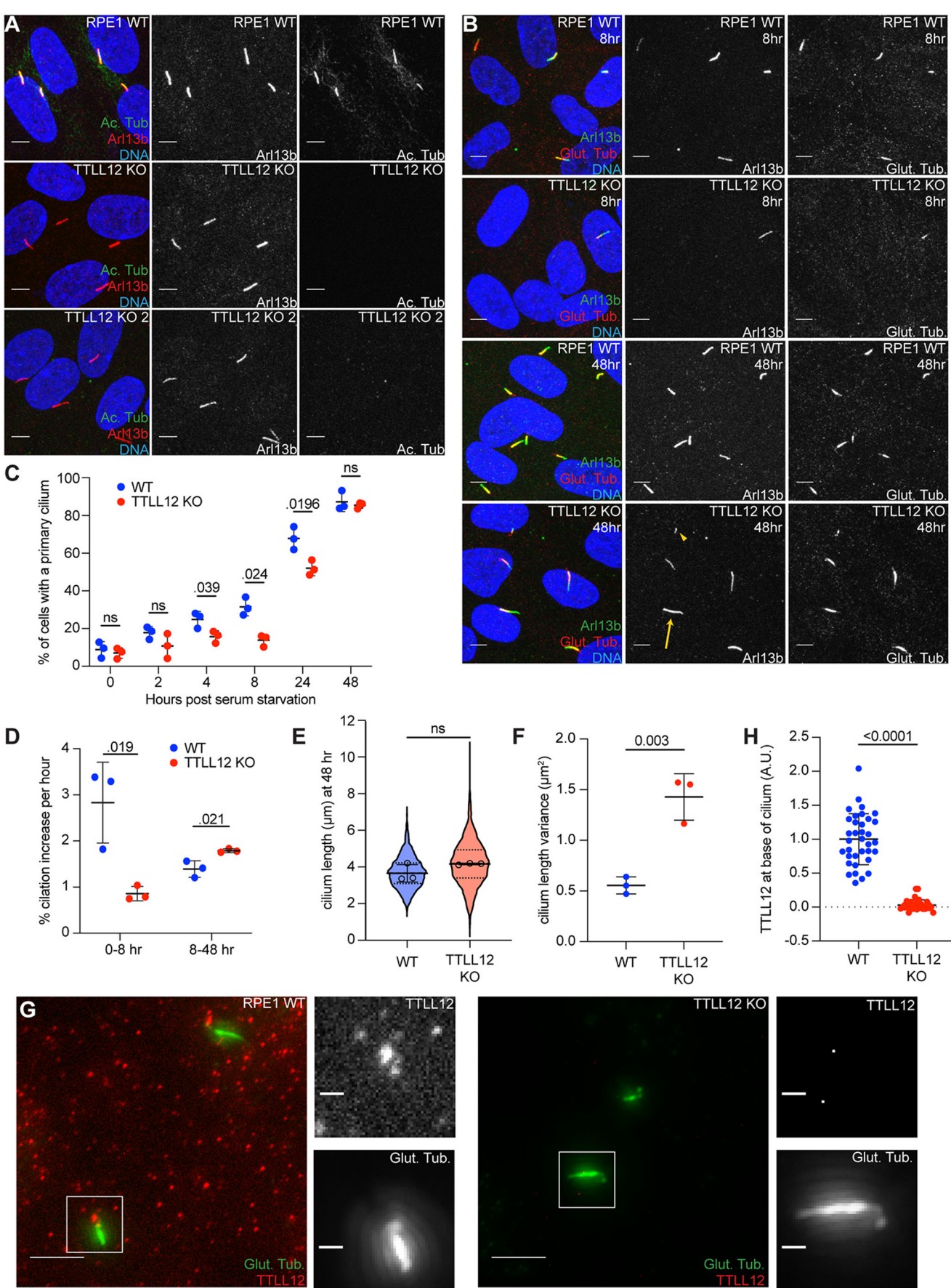

**Figure 4. TTLL12 regulates ciliation timing and length in RPE1 cells.**

(A) Representative images of RPE1 WT and TTLL12 KO cells grown on collagen-coated glass coverslips, serum-starved for 48 h, and stained for DNA, Arl13b, and acetylated tubulin (Ac. Tub). Scale bar = 5 μm. (B) Representative images of RPE1 WT and TTLL12 KO cells serum-starved for 8 or 48 h and stained for DNA, Arl13b, and GT335 (Glut. Tub.). Scale bar = 5 μm. Yellow arrowhead indicates a shorter than average cilium, and yellow arrow indicates a longer than average cilium. (C) Quantification of ciliation frequency through a serum starvation time course using Arl13b and GT335 as markers for cilia. The graph shows mean ± SD derived from three independent experiments. Student's t test (two-tailed) was used for statistical analysis. (D) Quantification of the rate of ciliation from the time course in (C). The graph shows mean ± SD derived from three independent experiments. Student's t test (two-tailed) was used for statistical analysis. (E) Quantification of primary cilium length after 48 h of serum starvation. The violin plot represents all primary cilia measured. Black circles represent average cilium length from three independent experiments, on which a t test was performed ($N = 3$, $n = 1438$ for WT, $n = 1352$ for KO). (F) Quantification of the length variance from three independent experiments. The graph shows mean ± SD derived from three independent experiments. Student's t test (two-tailed) was used for statistical analysis. (G) Representative images of TTLL12 localization in ciliated RPE1 WT and TTLL12 KO cells. Scale bar = 5 μm. Scale bar = 1 μm in inset. (H) Quantification of TTLL12 at the base of the primary cilium from (G). $N = 1$ and 35 cells were measured in each condition. Graph shows mean ± SD derived from three independent experiments. Student's t test (two-tailed) was used for statistical analysis. Source data are available online for this figure.

inhibitor, tubastatin A, and asked whether this treatment could rescue the effects of TTLL12 KO on microtubule acetylation. Again, we see tubastatin A treatment increased the amount of acetylated tubulin in WT, but not in TTLL12 KO cells (Fig. EV4B). Together, this data suggests tubulin is not being acetylated in TTLL12 KO cells, even though αTAT1 is expressed similarly in WT and TTLL12 KO cells (Fig. EV4C).

Previously, axoneme acetylation has been described as necessary for rapid ciliation initiation (Shida et al, 2010). To understand if TTLL12 functions in other aspects of axoneme formation, such as ciliation initiation, we performed a ciliation time course, where the number of ciliated cells was quantitated at 0, 2, 4, 8, 24, and 48 h after serum starvation (Fig. 4C). We observed a decrease in ciliation in TTLL12 KO cells at 4, 8, and 24 h, but this recovers by 48 h (Fig. 4B,C). It has previously been shown that RPE1 cells display two rates of ciliation, a fast phase from 0 to 8 h and a slow phase from 8 to 48 h (Jewett et al, 2023), and our analysis identifies two distinct rates as well (Fig. 4D). Interestingly, this is reversed in TTLL12 KO cells, which ciliate about 3× slower than WT from 0 to 8 h and slightly faster than WT from 8 to 48 h (Fig. 4D). This suggests ciliogenesis is delayed in TTLL12 KO cells, which is consistent with the lack of axoneme acetylation and the similar delay was observed when αTAT1 is knocked down (Shida et al, 2010).

The transition zone (TZ) is the boundary between the axoneme and basal body that restricts access to and from the cilium and is required for proper cilia function. To determine if TTLL12 KO affects transition zone formation, we compared the localization of 3 TZ proteins, NPHP4, RPGRIP1L, and TMEM67 in WT and TTLL12 KO RPE1 cells. We observed no difference in the TZ protein localization in TTLL12 KO cells (Fig. EV4D,E), which suggests TTLL12 does not affect the formation of the TZ. Additionally, we sought to understand if cilia length was altered in the absence of TTLL12. Primary cilium lengths were measured at the 8- and 48-h time points. While cilia frequency was decreased at 8 h, the average cilium length was slightly shorter but not significantly different in TTLL12 KO cells (EV Fig. 4F). At 48 h, the average cilium length in TTLL12 KO cells was slightly longer but also not significantly different than WT (Fig. 4E). However, we did observe a significantly larger variation in cilia length in TTLL12 KO cells (Fig. 4B, yellow arrows, F), which suggests TTLL12 may be required for either establishing or maintaining constant cilia length.

Finally, we asked if TTLL12 localizes to primary cilia. To test this, we generated a TTLL12 polyclonal antibody using GST-TTLL12 (aa1-260) as an antigen. We stained both WT and TTLL12 KO RPE1 cells for TTLL12 and glutamylated tubulin to mark cilia (Fig. 4G). We also pre-permeabilized cells before fixation since TTLL12 is generally cytosolic, similar to other TTLLs. We found TTLL12 localizes to the base of the primary cilium, and this signal is absent in TTLL12 KO cells (Fig. 4G,H), which supports TTLL12 functioning in primary ciliogenesis. Furthermore, we then asked if this TTLL12 localization was dependent on Rab19. To determine this, we stained Rab19 KO RPE1 cells (Jewett et al, 2021) for TTLL12 and measured TTLL12 intensity at the base of the primary cilium, presumably basal body (Fig. EV3G). We observed no change to TTLL12 localization in Rab19 KO cells (Fig. EV4H), which indicates Rab19 is not required for TTLL12 targeting to the basal body. Overall, we find TTLL12 is not required for ciliation in nonpolarized cells utilizing the intracellular ciliogenesis pathway, however, TTLL12 is required for acetylation of microtubules, an efficient ciliation rate, and regulating ciliary length. Furthermore, TTLL12 localizes to the base of the primary cilium near the basal body, and this localization is not dependent on Rab19.

## TTLL12 is required for axoneme extension in polarized epithelial cells

Since there are several key differences in ciliation among nonpolarized and polarized cells, we next sought to understand whether TTLL12 functions in extracellular ciliogenesis, which is common in polarized epithelial cells, such as MDCKs. To examine if TTLL12 is important for ciliation in polarized epithelial cells, MDCK WT and TTLL12 KO cells were grown on transwell filters for 8 days to allow for cells to fully polarize, and then fixed and stained for various ciliary markers. First, we looked at the axonemal microtubule marker acetylated tubulin (Fig. 5A,B), which begins to accumulate around day 3–4 (Jewett et al, 2021). Interestingly, while TTLL12 KO had no effect on clearing of apical cortical actin at the basal body docking site, it led to a decrease in acetylated axonemal microtubules in TTLL12 KO cells (Fig. 5A–C), and this decrease could be rescued by overexpressing GFP-TTLL12 (Figs. 5C and EV1E). Importantly, the decrease in MDCK cell ciliation was not due to an increase in proliferation (Fig. EV2G). Interestingly, GFP-TTLL12 overexpression appears to increase the length of the cilia (Fig. 5C). Thus, it is tempting to speculate that TTLL12 may regulate microtubule stability and/or growth during axonemal extension, although further studies will be needed to determine that. These data suggest that TTLL12 is not necessary for

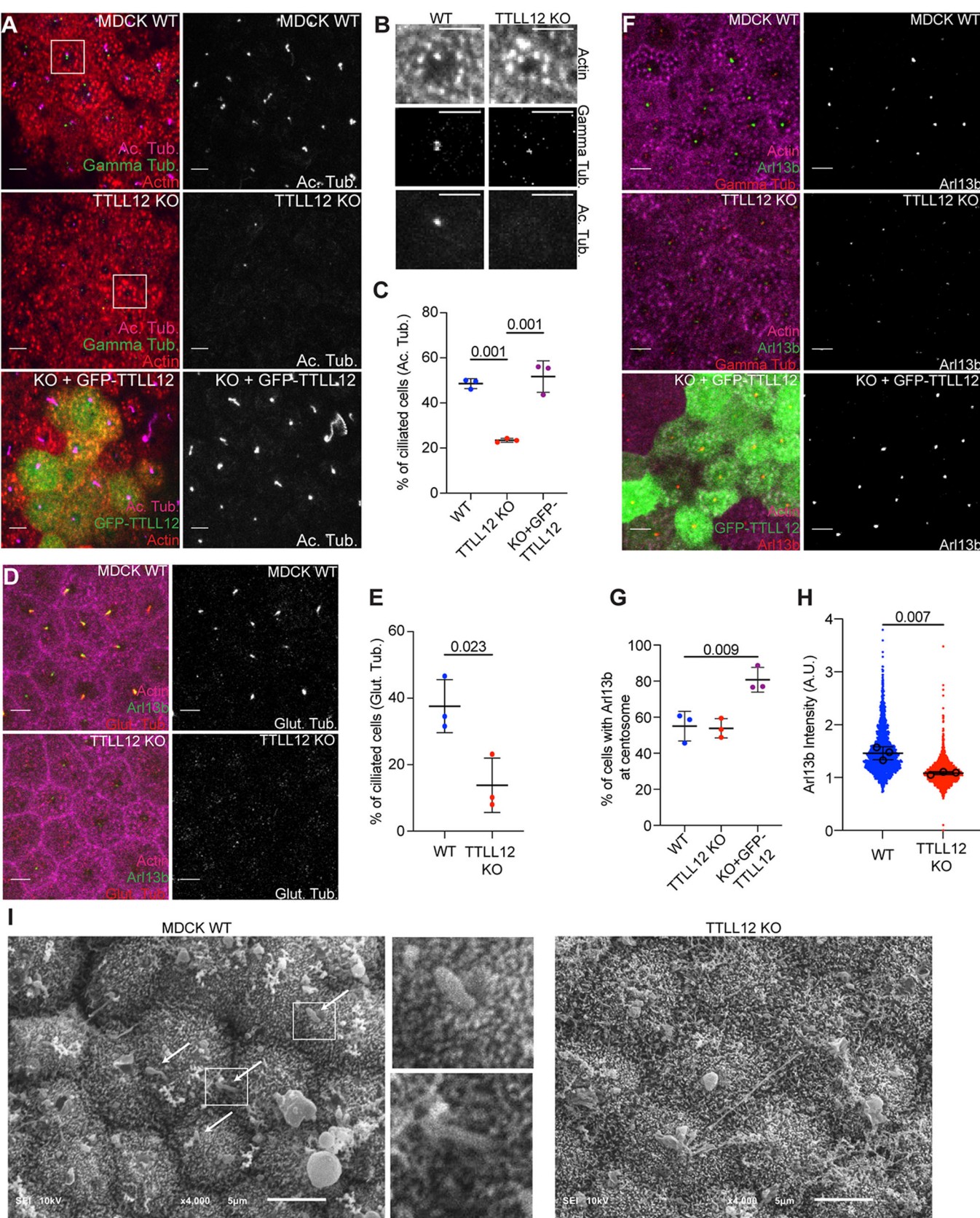

◄

**Figure 5.   TTLL12 is required for axoneme formation in polarized epithelial cells.**

(A) Representative images of MDCK WT, TTLL12 KO, and TTLL12 KO cells stably expressing GFP-TTLL12 grown on transwell filters, polarized and stained for actin, gamma-tubulin, and acetylated tubulin (Ac. Tub). Scale bar = 5 μm. (B) Inset from A (white boxes). Scale bar = 5 μm. (C) Quantification of cells with a primary cilium marked by acetylated tubulin in (A). Graph shows mean ± SD from three independent experiments. Student's $t$ test (two-tailed) was used for statistical analysis. (D) Representative images of MDCK WT and TTLL12 KO cells stained for actin, Arl13b, and GT335. Scale bar = 5 μm. (E) Quantification of cells with a primary cilium marked by glutamylated tubulin (GT335). Graph shows mean ± SD from three independent experiments. Student's $t$ test (two-tailed) was used for statistical analysis. (F) Representative images of MDCK cells stained for actin, gamma-tubulin, and Arl13b. Scale bar = 5 μm. (G) Quantification of cells with Arl13b at the centrosome (marked by gamma tub.). Graph shows mean ± SD from three independent experiments. Student's $t$ test (two-tailed) was used for statistical analysis. (H) Quantification of Arl13b intensity at each gamma-tubulin puncta. Black circles represent average Arl13b intensity from three independent experiments ($N = 3$, $n = 1658$ for WT, $n = 1616$ for KO). Graph shows mean ± SD. Student's $t$ test (two-tailed) was used for statistical analysis. (I) SEM images of the apical surface of MDCK WT and TTLL12 KO polarized on transwell filters. Images are at ×4000 magnification with 5 μm scale bar. White arrows indicate primary cilia-like structures. Boxes mark the location of higher magnification insets. Source data are available online for this figure.

centrosome migration or apical cortical actin clearing but may be required for regulating axoneme formation or maintenance. However, the lack of acetylated axonemal microtubules could be explained by two possibilities. First, there could be a failure to extend axonemal microtubules after basal body docking. Second, similar to our observation in RPE1 cells, MDCK cells may still form an axoneme, but not acquire axoneme acetylation. Therefore, we next looked for axonemal microtubules using the glutamylated tubulin marker, GT335 (Fig. 5D). Interestingly, we found there is a loss of ciliary axoneme glutamylation in TTLL12 KO cells (Fig. 5E), which suggests that TTTLL12 depletion may result in a complete lack of axoneme formation.

To further dissect the role of TTLL12 in epithelia ciliation, we next analyzed the ciliary membrane marker Arl13b (Fig. 5F), which in polarized MDCK cells arrives at the centrosome around day 2–3 and increases as axoneme growth occurs (Jewett et al, 2021). We observed that there are the same number of cells with Arl13b at the centrosome (marked by gamma-tubulin) in WT and TTLL12 KO cells (Fig. 5G). However, we also observe that these Arl13b puncta are significantly smaller in TTLL12 KO cells (Fig. 5H), which suggests that TTLL12 is not required for initial ciliary membrane recruitment to the centrosome but may be necessary for ciliary membrane accumulation associated with axoneme extension. To better assess the presence of extracellular axonemes in these MDCK cells, we next performed scanning electron microscopy of the apical cell surface. We were able to detect ciliary-like structures in WT but not TTLL12 KO cells (Fig. 5I, white arrows). Altogether, our data suggest there is a lack of axonemal microtubules, and thus cilia, in TTLL12 KO MDCK cells, suggesting that TTLL12 is an important factor for ciliogenesis in polarized epithelial cells.

## TTLL12 regulates mitotic spindle positioning and cell migration

Our data so far demonstrates that TTLL12 regulates microtubule stability and is required for cilia formation in epithelial cells. Interestingly, in addition to the basal body, TTLL12 also appears to associate with mitotic spindle centrosomes where it affects mitotic microtubule stability (Fig. 3). Based on these data, we wondered whether, in addition to cilia, TTLL12 may also be required for mitotic spindle formation and/or positioning. Indeed, mitotic spindle positioning in polarized epithelial cells plays a key role in the formation of various epithelial structures, such as expansion of a single apical lumen. To test whether TTLL12 may play a role during mitotic spindle formation, we fixed MDCK WT and MDCK TTLL12 KO cells and stained them using anti-α-tubulin antibodies.

Cells were then analyzed for the ability to form and position bipolar mitotic spindles. As shown in Fig. 6A, TTLL12 KO cells could form bipolar mitotic spindles and did not appear to have any issues in chromosome segregation. However, we did observe defects in mitotic spindle positioning (Fig. 6A–C). Typically, in dividing MDCK cells, the mitotic spindle is positioned in the center of the cells and is aligned with the major axis of the cell (Fig. 6A). Depletion of TTLL12 resulted in defects in centering of the mitotic spindle (Fig. 6B), as well as aligning the spindle with the major axis of the cell.

Microtubules also play an important role in directional cell migration by forming a microtubule array from the centrosome to the leading edge of the cell. This microtubule array is presumably needed for the delivery of various migration-mediating proteins, such as integrins. The key to the function of this leading-edge microtubule array is its high assembly-disassembly dynamics which allows for cells to rapidly move and change directions. Indeed, recent studies suggested that decreases in microtubule acetylation increases cell migration (Hubbert et al, 2002; Palazzo et al, 2003). Since TTLL12 KO reduces tubulin acetylation in RPE1 cells, we also decided to explore whether TTLL12 may regulate cell migration. First, we co-stained RPE1 cells with anti-TTLL12 and anti-α-tubulin antibodies. Cells were plated at low density and incubated with serum-supplemented media to allow migration and formation of lamellipodia. Consistent with our previous data, most of the TTLL12 was present in the cytosol (Fig. 6D). However, a subset of TTLL12 was clearly enriched at the edge of the lamellipodia, suggesting that TTLL12 may also be involved in regulating lamellipodia-associated microtubule assembly, and consequently cell migration. Indeed, scratch wound migration assay demonstrate that RPE1 TTLL12 KO cells migrate faster than RPE WT cells (Fig. 6E,F). This is consistent with previous studies showing a decrease in microtubule acetylation increases cell migration. Interestingly, TTLL12 KO cells clearly express αTAT1 (see Fig. EV4C), thus, how TTLL12 inhibit microtubule acetylation remains to be determined. Taken together, our data suggest that TTLL12 plays a role in several microtubule-dependent processes, although further studies will be needed to better understand the roles of TTLL12 during cell migration and mitotic spindle positioning.

## TTLL12 regulates tubulin methylation

Our data so far clearly demonstrate that TTLL12 is required for the formation and positioning of variety of microtubule structures, including cilia and mitotic spindle. We also show that TTLL12

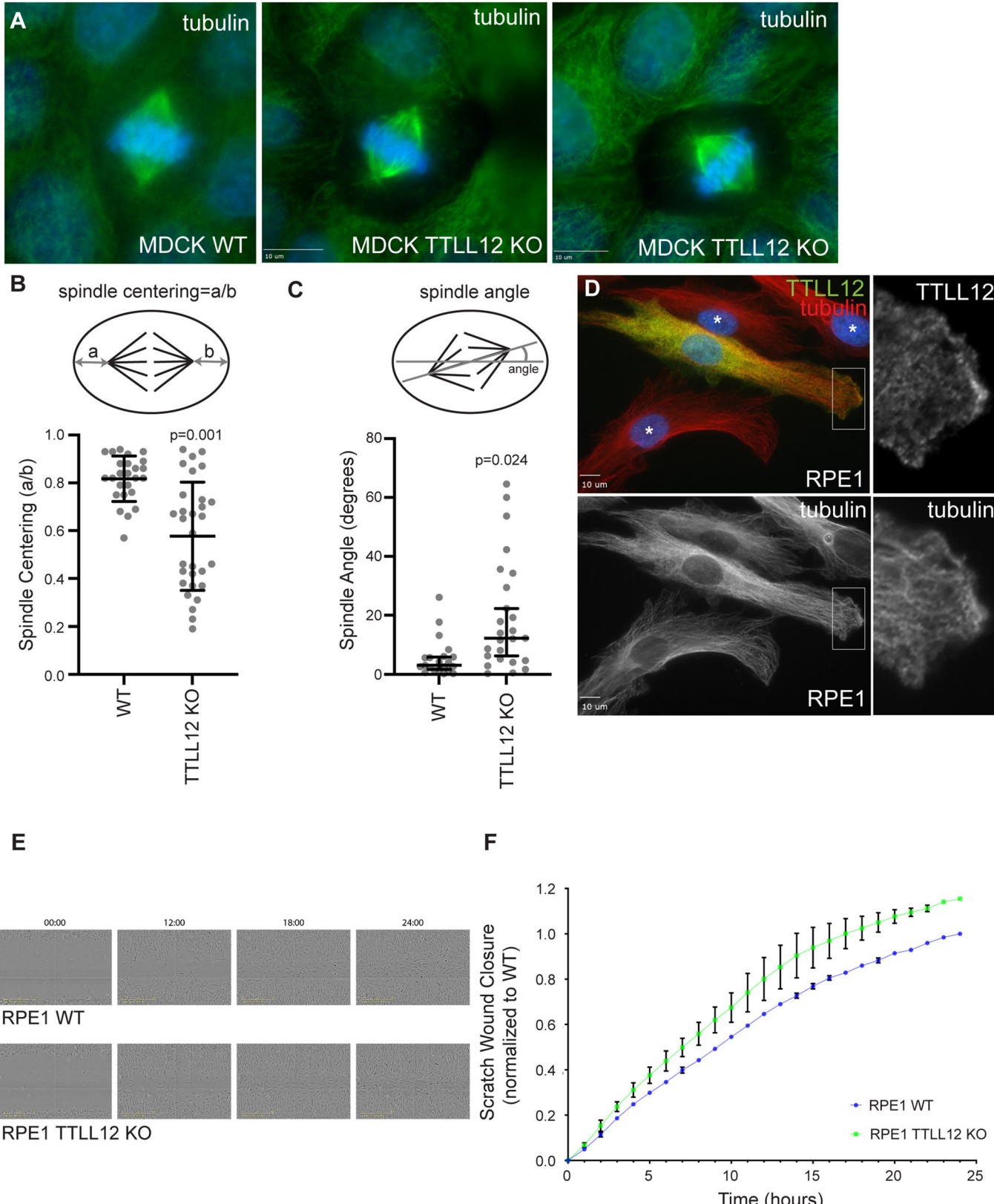

**Figure 6. TTLL12 regulates cell migration and mitotic spindle positioning.**

(A) Representative images of mitotic MDCK WT and TTLL12 KO cells grown on collagen-coated glass coverslips and stained for tubulin. Scale bar = 10 μm. (B) Quantification of mitotic spindle centering in MDCK WT and TTLL12 KO cells. Each circles represents measurement derived from single cell. The data shown are the means and standard deviations derived from three independent experiments. Student's *t* test (two-tailed) was used for statistical analysis. (C) Quantification of mitotic spindle angle in MDCK WT and TTLL12 KO cells. Each circles represents measurement derived from single cell. The data shown are the means and standard deviations derived from three independent experiments. The data shown are the median and 95% confidence interval. Student's *t* test (two-tailed) was used for statistical analysis. (D) Representative images of mixed culture of RPE1 WT and TTLL12 KO (marked with asterisk) cells grown on collagen-coated glass coverslips and stained for tubulin and TTLL12. Box marks the area enlarged in insets on the right. Scale bar = 10 μm. (E) Representative time-lapse still images of RPE1 WT and RPE1 TTLL12 KO migrating into scratch wound. (F) Quantification of scratch wound migration analysis of RPE1 WT and RPE1 TTLL12 KO performed using Incucyte. Cells were analyzed for 24 h with 1-h time-lapse. The relative wound density was calculated for each time point and normalized against last time point for RPE1 WT cells. The data shown is derived from one independent experiment. Error bars represent variability between three different technical repeats. Source data are available online for this figure.

regulates microtubule stability, dynamics, and acetylation. What remains unclear, however, is how TTLL12 performs all these functions. A previous report suggests TTLL12 does not have an enzymatically active TTL domain and identified a putative SET-like domain in the TTLL12 N-terminus (aa 91–259) (Brants et al, 2012). However, whether this SET-like domain is functionally active remains unclear, especially since it is quite divergent from canonical SET domains. To better visualize the SET and TTL domains in TTLL12, we used Alpha-Fold to analyze the predicted structure of TTLL12. This structure prediction suggests that the TTL domain and SET domain are in distinct regions and seemingly form two halves of TTLL12 that are located adjacent to each other (Fig. 7A). SET domains are classically contained in proteins that methylate histone lysines (Dillon et al, 2005), and several recent studies have proposed there are certain histone methyltransferases that also methylate α-tubulin at K40 (K40me3), K311 (K311me1), and K394 (K394me3) (Chin et al, 2020; Li et al, 2020; Park et al, 2016). Therefore, we hypothesized TTLL12 may be another SET-containing protein responsible for methylating tubulin.

To further confirm that tubulin can be methylated, we performed mass spectrometry on purified porcine microtubules. Consistent with previous reports, we did find that in porcine brain microtubules, both α- and β-tubulins are methylated on several lysine and arginine residues, including α-tubulin R2me2, K40me3, R79me1, K124me3, and K166me3 and β-tubulin R2me2, R86me2, and R318me1 and 1–10% of peptides are methylated at each residue, except for α-tubulin K124, which is always tri-methylated (see Dataset EV2).

Porcine tubulin is purified from brain and it contains predominately axonal tubulin, which is known to be heavily modified. That, of course, does not mean that ciliary tubulin has the same PTMs, especially methylation. To start testing whether ciliary tubulin is methylated, we took advantage of the cilia purification protocol that was developed by Dr. Wallace Marshall's laboratory (Ishikawa et al, 2013). This protocol uses buffer with high calcium concentration to de-ciliate mIMCD-3 cells. Cilia collected in deciliation buffer can then be purified using series of gradient centrifugations (Ishikawa et al, 2013). We found that high calcium also de-ciliates MDCK cells (Fig. EV5F), thus, we used Marshall's protocol to purify cilia from MDCK cells. Importantly, blotting purified MDCK cilia with various methylation antibodies have suggested that ciliary tubulin may be mono-methylated on lysine, as well as mono- and di-methylated on arginine. These data are fully constant with our hypothesis that tubulin methylation regulates cilia formation and/or stability. Further studies, however,

will be needed to unequivocally show methylation of axonemal tubulin.

To attempt to identify the tubulin sites that may be methylated in TTLL12-dependent fashion, we performed mass spectrometry after incubating purified porcine brain tubulin with purified 6His-TTLL12 and methyl donor S-adenosyl-L-methionine (SAM) for 30 min at 37 °C. This type of analysis did not reveal an obvious increase in methylation at any specific residue of α- or β-tubulin except for α-tubulin-R79 (Dataset EV2). However, further studies will be needed to determine whether TTLL12 targets α-tubulin-R79 or any other specific tubulin sites for methylation.

To test the hypothesis that TTLL12 may function as tubulin methyltransferase, we used a fluorescence-based methyltransferase assay. In the assay, an enzyme is incubated with a substrate and the SAM. As methylation occurs and SAM is converted to S-adenosylhomocysteine (SAH), SAH is converted to a fluorescent compound, providing a way to measure methylation over time in vitro. First, we observe very little fluorescence accumulation with purified porcine brain tubulin alone (Fig. 7B), which is expected given that there is no enzyme present. However, we do observe an increase in fluorescence with TTLL12 alone (Fig. 7B). It is common for methyltransferases to auto-methylate, suggesting that TTLL12 is potentially an active methyltransferase. When TTLL12 is incubated with brain tubulin, we see an additional increase in fluorescence (Figs. 7B and EV5A), which suggests that tubulin may be a substrate of TTLL12. Importantly, consistent with a previous report that TTLL12 is not a histone methyltransferase (Brants et al, 2012), there was no increase in fluorescence when TTLL12 was incubated with histone H3 (Fig. EV4B). In addition, we tested GST-TTLL12 (aa1–260), which contains the SET but not TTL domain, and found no effect on tubulin methylation (Fig. EV5C), suggesting that TTLL12 needs both, SET and TTL domains, to exert its putative tubulin methyltransferase activity. Since fluorescence is directly related to the amount of SAH produced, we can then use a standard curve of the fluorescent compound to determine the enzymatic rate. The rates were calculated using the initial velocity from the first 5 min (Fig. 7B, dotted box) and tubulin alone was subtracted as background (Fig. 7C). We observed a concentration-dependent increase in TTLL12 methyltransferase activity when combined with tubulin (Fig. 7C) and see no activity with the TTLL12-SET domain alone. This suggests that the TTL domain is required for both, TTLL12 enzymatic activity and binding to tubulin. Taken all together, this data suggests TTLL12 is a putative tubulin methyltransferase, at least in vitro, and this activity requires the TTL domain.

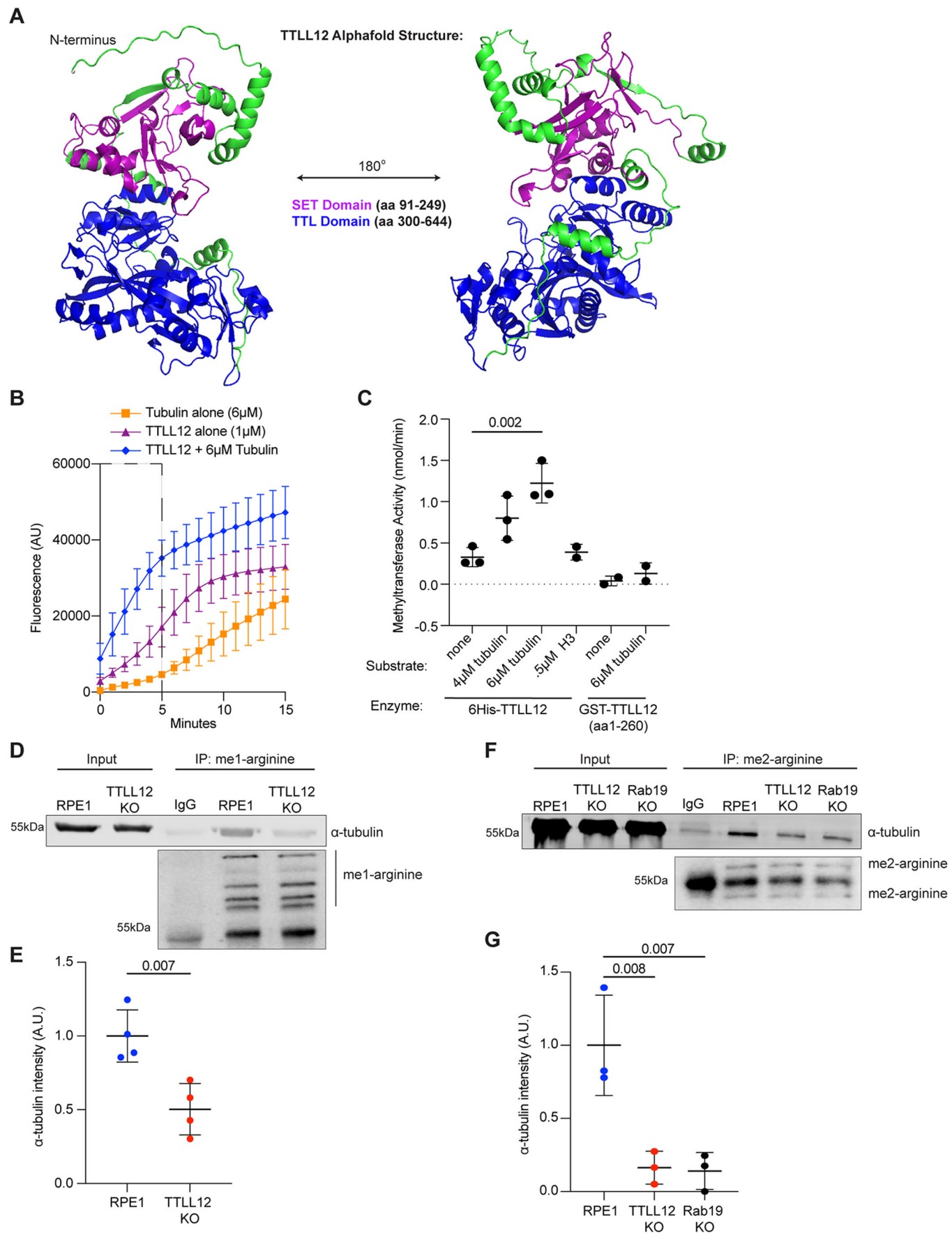

**Figure 7. TTLL12 regulates tubulin methylation.**

(A) Alpha-fold structure of TTLL12. SET domain is purple. TTL domain is blue. Amino acids not in defined domains are green. (B) Fluorescence-based assay to measure methyltransferase activity. In total, 1 μM 6His-TTLL12 was incubated with methyl donor SAM and 6 μM porcine brain tubulin and fluorescence was measured over time. The data are the means and standard deviations derived from three different experiments. Mass spectrometry identifying methylation of tubulin alone and TTLL12 + tubulin is in Dataset EV2. (C) Methyltransferase activity was calculated using the slopes of the curves from C (inside the dotted box). The data shown are the means and standard deviations derived from three different experiments. (D) Immunoprecipitation of mono-methyl arginine from RPE1 cells followed by western blot for α-tubulin. Left columns show lysates (input) probed for Rme1 and α-tubulin. Right columns show immunoprecipitates probed for Rme1 and α-tubulin. (E) Quantification of α-tubulin band from (D). IgG control was subtracted as background. Graph shows mean ± SD from four independent experiments. Student's *t* test (two-tailed) was used for statistical analysis. (F) Immunoprecipitation of di-methyl arginine from RPE1 cells followed by western blot for α-tubulin. Left column shows lysates (input) probed for Rme2 and α-tubulin. Right columns show immunoprecipitates probed for Rme2 and α-tubulin. (G) Quantification of α-tubulin band from (F). IgG control was subtracted as background. Graph shows mean ± SD from three independent experiments. Student's *t* test (two-tailed) was used for statistical analysis. Source data are available online for this figure.

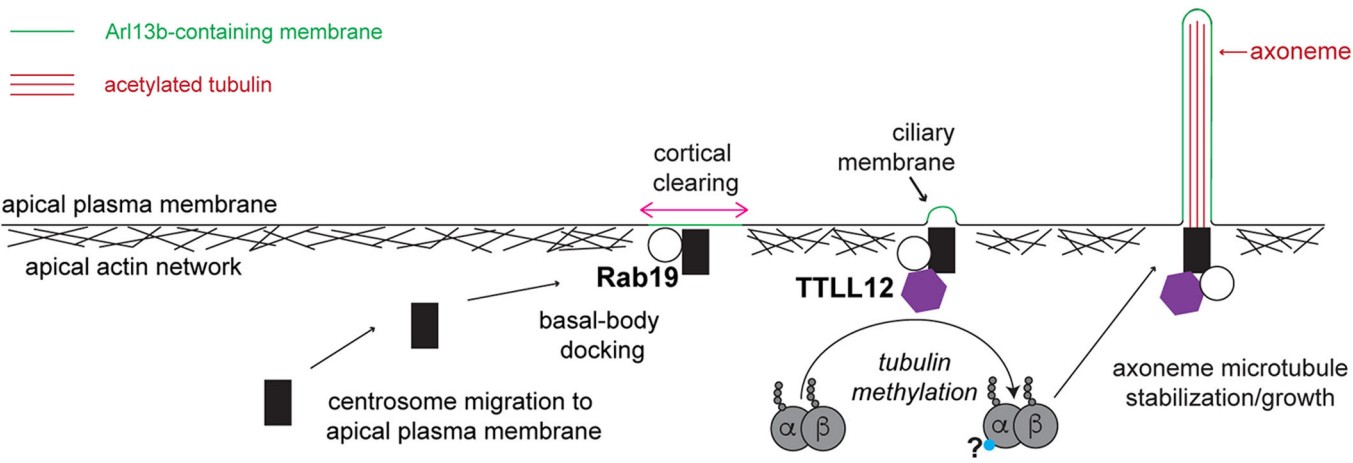

**Figure 8. Model figure.**

In polarized epithelial cells, the centrosome migrates to the apical surface, and Rab19 is required for apical actin clearing and initial ciliary membrane recruitment. TTLL12 also resides at the centrosome and interacts with the α/β-tubulin heterodimer to promote tubulin arginine methylation and lysine acetylation, which is required for axoneme growth and stability.

Since we were able to identify the presence of methylation on tubulin, we next asked if tubulin methylation occurs in cells and whether TTLL12 affects this process. To test this, we immunoprecipitated methylated proteins from RPE1 WT and TTLL12 KO cells using antibodies to different methyl marks (Kme1, Kme3, Rme1, or Rme2) and probed for α-tubulin. Lysine mono-methylation of α-tubulin was not detectable by this method (Fig. EV5D), and lysine tri-methylation of α-tubulin was faintly detectable but unchanged in TTLL12 KO cells (Fig. EV5E), which suggests that TTLL12 does not affect tubulin lysine methylation. Surprisingly, we found a significant decrease in arginine mono-methylation of α-tubulin in TTLL12 KO cells (Fig. 7D,E), which suggests that TTLL12 may promote tubulin arginine methylation. Since many methyltransferases can catalyze both, mono- and di-methylation, we then measured arginine di-methylation of α-tubulin (Fig. 7F) and again observed a significant decrease in TTLL12 KO cells (Fig. 7G). In addition, we observed this same loss of α-tubulin arginine di-methylation in Rab19 KO cells, which suggests that either Rab19 is necessary for TTLL12 activity, or Rab19 and TTLL12 both function in processes that similarly affect α-tubulin arginine di-methylation. Together, this data indicates that in addition to α-tubulin acetylation,

TTLL12 also regulates, directly or indirectly, α-tubulin arginine methylation.

Overall, we find TTLL12 is a novel Rab19-binding partner required for ciliogenesis in polarized epithelial cells. We identify TTLL12 as a regulator of tubulin PTMs that interacts with the α/β-tubulin heterodimer and promotes microtubule stability. Our data suggest a model where Rab19 functions in cortical clearing and initial ciliary membrane recruitment, but also interacts with TTLL12 at the centrosome, where TTLL12 is necessary for the growth and stabilization of axonemal microtubules (Fig. 8).

## Discussion

The primary cilium is a vital microtubule-based structure that protrudes from the cell surface and responds to the external environment. Defects in primary cilia formation or function are associated with a multitude of diseases and can commonly affect the renal system. Renal cells are polarized epithelial cells that utilize the extracellular ciliogenesis pathway, which has been understudied in comparison to the intracellular ciliogenesis pathway. However, we do know that in the extracellular

ciliogenesis pathway, the centrosome migrates to the apical surface, which coincides with a clearing in the apical cortical actin and is followed by axoneme extension into the extracellular space. This process is tightly coupled to epithelial cell polarization, and while we and others have previously characterized the timing of ciliogenesis relative to polarization, it remains unclear what factors are regulating axoneme growth and stability at the apical plasma membrane. In this study, we identified TTLL12 as a novel regulator of axoneme extension in polarized epithelial cells. We propose that TTLL12 actively functions at the basal body and interacts with the α/β-tubulin heterodimer before incorporation into the primary cilium. This interaction promote microtubule stability and regulates (likely indirectly) the tubulin acetylation required for extension and maintenance of the axoneme (Fig. 8). TTLL12 is an exciting candidate for this role, as many TTLLs have been found to regulate cilia formation and function, but TTLL12 has not previously been investigated. Our work suggests TTLL12 is necessary for primary cilia formation in polarized epithelia cells and begins to dissect the mechanism of this mysterious TTLL.

## The complicated link between TTLL12 and microtubule PTMs/dynamics

Here we find that TTLL12 directly binds the α/β-tubulin heterodimer, which is unusual for a TTLL, as all other TTLLs preferentially bind the microtubule polymer. However, we also find that TTLL12 promotes microtubule stability and affects microtubule polymerization. This is interesting given the lack of interaction between TTLL12 and the microtubule polymer and suggests that the interaction between TTLL12 and the tubulin heterodimer is somehow influencing microtubules in the polymer state. While further work will be necessary to determine the molecular mechanism of this influence, it is tempting to speculate that TTLL12 is altering tubulin PTMs either directly (enzymatically) or indirectly (in a complex) and that these PTMs are affecting the microtubule polymer and/or microtubule-binding partners. Alternatively, TTLL12 could bind a certain sub-population of the tubulin heterodimer and preventing the incorporation of these specific subunits into the microtubule polymer.

Interestingly, TTLL12 affects multiple tubulin PTMs, as we found TTLL12 promotes both α-tubulin K40 acetylation and arginine methylation. This presents multiple possibilities: either TTLL12 affects one specific PTM, and PTMs influence one another or TTLL12 regulates a population of microtubules that acquire multiple PTMs, so if this population is lost, multiple PTMs are affected. In RPE1 cells, neither stabilizing microtubules with Taxol nor treating with an HDAC6 inhibitor restored microtubule acetylation. In MDCK cells, GFP-TTLL12 rescued cilia formation in TTLL12 KO cells, but not overall acetylation levels. Microtubule acetylation is catalyzed by αTAT1, which is thought to be the major, and probably only, α-tubulin K40 acetyltransferase (Shida et al, 2010). K40 acetylation is associated with more stable, longer-lived microtubules; however, K40 acetylation is not required for cell viability or organism development (Akella et al, 2010; Kalebic et al, 2013; Shida et al, 2010). Therefore, it is possible that TTLL12 indirectly promotes K40 acetylation, potentially by indirectly regulating αTAT1 ability to bind and/or acetylate K40

in polymerized microtubules. It is tempting to speculate that this αTAT1 activity may be regulated by methylating α/β-tubulin heterodimer (see discussion below), which later gets incorporated into axonemal microtubules. In addition, this indirect mechanism may also explain the discrepancies we observe in acetylation loss between MDCK and RPE1 cells. Polarized MDCK cells have a population of stable non-centrosomal microtubules that run from apical to basolateral poles of the cell (Bre et al, 1987). Therefore, it is possible TTLL12 does not regulate the stability/acetylation of this population of microtubules, and these microtubules are able to retain acetylation in MDCK TTLL12 KO cells. This may suggest TTLL12 activity is limited to the basal body, and while Rab19 is not required for TTLL12 targeting to the basal body, it will be interesting to determine if Rab19 binding is involved in regulating basal body-associated TTLL12 activity. Indeed, our data demonstrates that TTLL12 can co-bind Rab19 and tubulin heterodimer simultaneously. Thus, it is tempting to speculate that Rab19 may stimulate presumptive TTLL12 methyl-transferase activity. Overall, further work will be necessary to delineate how TTLL12 directly affects tubulin acetylation that influences microtubule dynamics or if TTLL12 is affecting microtubule dynamics that consequently change tubulin acetylation. Additionally, further studies will be needed to define the biochemical properties and functional significance of TTLL12 and Rab19 interaction.

## Differential requirement for TTLL12 in primary ciliogenesis

We have identified that TTLL12 localizes to the basal body and functions in primary cilia formation. Importantly, TTLL12 is required for ciliogenesis, specifically axoneme formation and/or maintenance in polarized epithelial cells. However, loss of TTLL12 only delays ciliation and ultimately is not required for ciliogenesis in nonpolarized cells. This discrepancy may stem from the fact that these cells utilize different ciliogenesis pathways. In polarized epithelial cells, such as renal MDCK cells, the centrosome migrates and docks at the apical plasma membrane before axoneme growth directly into the extracellular space. In contrast, in nonpolarized cells, such as RPE1 cells, axoneme growth begins in the cytoplasm inside the ciliary vesicle before centrosome migration and ciliary vesicle fusion with the plasma membrane. Because of this process, the axoneme in nonpolarized cells becomes only partially exposed to the extracellular space, and the primary cilium tends to sit in a ciliary pocket. In RPE1 cells, it has been found that this pocket is deep, and the primary cilium is almost completely submerged (Ghossoub et al, 2011). Consequently, in RPE1 cells, the growing axoneme is protected by the ciliary vesicle, and the developed axoneme is protected by the ciliary pocket. Consistent with our results, this again suggests a model where TTLL12 promotes axonemal microtubule stability. Consequently, intracellular ciliogenesis in RPE1 cells may be delayed, but eventually able to compensate for the loss of stability provided by TTLL12, and TTLL12 function may be more critical in polarized epithelial cells, where the growing axoneme is less protected. This work suggests polarized epithelial cells have additional requirements for axoneme extension/stability and it will be interesting in the future to identify how nonpolarized cells are able to compensate for the loss of TTLL12.

## Potential enzymatic function of TTLL12

TTLL12 is the only member of the TTLL family without a clearly defined enzymatic function, which is what makes it difficult to determine if TTLL12 is directly affecting a specific tubulin PTM. Since we demonstrate that TTLL12 interacts with tubulin heterodimer and regulates microtubule dynamics, it raises the interesting question of whether TTLL12 is a tubulin-modifying enzyme. Although it is in the TTLL family, overexpressing or depleting TTLL12 in cells does not affect tyrosination, glutamylation, or glycylation, and TTLL12 does not demonstrate tyrosinase, glutamylase, or glycylase activity in vitro (van Dijk et al, 2007; Rogowski et al, 2009; Brants et al, 2012). While TTLL12 does not demonstrate any known enzymatic activities of an active TTL domain, Brants et al (2012) previously identified a SET-like domain in the TTLL12 N-terminus. In addition, while SET domains are classically contained in proteins that methylate histone lysines (Dillon et al, 2005), the authors determined that TTLL12 did not methylate histones in vitro (Brants et al, 2012). More recently, several groups have identified that tubulin/microtubules can be methylated by several SET domain-containing proteins (Park et al, 2016; Chin et al, 2020; Li et al, 2020). Therefore, we hypothesized TTLL12 may be a tubulin-specific methyltransferase. In this study, we tested this hypothesis using a fluorometric methyltransferase assay and found an increase in fluorescence when tubulin was added to TTLL12. While this suggestive of TTLL12 being a tubulin-specific methyltransferase, our current results do not identify the specific site(s) of tubulin methylation by TTLL12. It is important to note that the porcine brain tubulin used in our experiments may not be the ideal substrate for TTLL12, since features of brain tubulin isotypes and/or pre-existing PTMs on this tubulin may impede methylation. More work will be needed to fully determine whether TTLL12 contains an enzymatically active SET domain and functions to regulate microtubule stability by methylating α/β-tubulin heterodimer.

To further pinpoint TTLL12 methyltransferase activity, we also investigated if TTLL12 affected tubulin methylation in cells. While we did not observe changes in tubulin lysine methylation, we did find that there was a decrease in tubulin arginine mono- and di-methylation in TTLL12 KO cells. However, arginine methylation is not typically catalyzed by canonical SET domain methyltransferases, but rather, by the PRMT family. It is possible that TTLL12 directly catalyzes this modification since TTLL12-SET-like domain is quite divergent from canonical SET domains. Alternatively, it is also possible that TTLL12 regulates tubulin arginine methylation indirectly by mediating lysine methylation of other, as of yet unidentified, tubulin methylating proteins. Arginine methylation on tubulin has not been extensively studied, so it will be exciting to determine how TTLL12 contributes to the regulation of tubulin arginine methylation, and the function(s) of this PTM. Interestingly, α-tubulin R2 is present in the α-tubulin and β-tubulin interaction interface, and it has been shown that R2N mutation in TUBA1A decreases microtubule stability, thus leading to developmental defects, known as tubulopathies (Gardner et al, 2018). On the other hand, α-tubulin-R79 is located at the microtubule lumen at close proximity of K40, tempting to speculate that methylation of R79 may affect αTAT1 ability to acetylate K40. These are very intriguing possibilities, however, more research in our and other laboratories will be needed to investigate this novel tubulin PTM.

In summary, our study has identified TTLL12 as a key regulator of ciliary axoneme extension in polarized renal epithelial cells. We have also shown that TTLL12 regulates microtubule stability, dynamics, acetylation, and methylation. While further studies will be needed to fully understand how TTLL12 functions, this work lays a foundation for future studies in understanding the regulation of cilia formation and stability in polarized epithelial cells.

# Methods

## Reagents and tools

See Table 1.

**Table 1. Reagents and tools.**

| Reagent/resource | Reference or source | Identifier or catalog number |
|---|---|---|
|  | Source (public): *Stock center, company, other labs* Reference: *list relevant study if referring to previously published work; use "this study" if new. If neither* applies: *briefly explain.* | *Provide catalog numbers, stock numbers, database IDs or accession numbers, RRIDs or other relevant identifiers.* |
| **Experimental models** *List cell lines, model organism strains, patient samples, isolated cell types etc. Indicate the species when appropriate.* |  |  |
| MDCK II (*Canis familiaris*) | ATCC | CRL-2936 |
| hTERT RPE1 (*H. Sapiens*) | ATCC | CRL-4000 |
| **Recombinant DNA** *Indicate species for genes and proteins when appropriate* |  |  |
| pFastBac HTA | Gift from Dr. Shaodong Dai |  |
| pLVX304-hTTLL12 | CCSB-Broad Lentiviral Expression Library | Clone Id: ccsbBroad304_02727 |
| pLVX-puro | Clontech | Cat #632164 |
| pGEX-6P1 | Gift from Dr. Rui Zhao |  |

**Table 1.** (continued)

| Reagent/resource | Reference or source | Identifier or catalog number |
|---|---|---|
| pCIG2-GFP-MACF18 | Gift from Dr. Jeff Moore | |
| **Antibodies** *Include the name of the antibody, the company (or lab) who supplied the antibody, the catalog or clone number, the host species in which the antibody was raised and mention whether the antibody is monoclonal or polyclonal. Please indicate the concentrations used for different experimental procedures.* | | |
| Mouse anti-α-tubulin clone B-5-1-2 | Sigma | Cat #T6074, 1:1000 |
| Rabbit anti-GFP | Invitrogen | Cat #A-11122, 1:1000 |
| Rabbit anti-GST | Gift from Dr. Andrew Peden | 1:1000 |
| Rabbit anti-Acetylated Tubulin | Cell Signaling | Cat #5335, 1:1000 |
| Mouse anti-Acetylated Tubulin | Sigma | Cat #T7451, 1:1000 |
| Mouse anti-β-Actin | Cell Signaling | Cat #3700S, 1:1000 |
| Rabbit anti-β-tubulin | Li-cor | Cat #926-42211, 1:1000 |
| Rabbit anti-polyglutamate chain | Adipogen | Cat #AG-25B-0030-C050, 1:1000 |
| Rat anti-tyrosinated Tubulin clone Y1/2 | EDM Millipore | Cat #MAB1864 |
| Mouse anti-Arl13b clone N295B/66 | Antibodies Inc. | Cat #73-287, 1:500 |
| Mouse anti-GT335 | Adipogen | Cat #AG-20B-0020-C100, 1:200 |
| Mouse anti-α-tubulin clone DM1α | Invitrogen | Cat #62204, 1:400 |
| Rabbit anti-TTLL12 | Prekeris Lab (this paper) | 1:500 WB, 1:400 IF |
| Mouse anti-gamma-tubulin | Sigma | Cat #T5326, 1:500 |
| Rabbit anti-RPGRIPL1 | Proteintech | Cat #55160-1-AP, 1:200 |
| Rabbit anti-NHP4 | Proteintech | Cat #13812-1-AP, 1:200 |
| Rabbit anti-TMEM67 | Proteintech | Cat #13975-1-AP, 1:200 |
| Rabbit anti-αTAT1 | | |
| Rabbit anti-di-methyl-arginine | Cell Signaling | Cat #13522S 1:1000 |
| Rabbit anti-mono-methyl-arginine | Cell Signaling | Cat #8015S 1:1000 |
| Rabbit anti-tri-methyl lysine | Cell Signaling | Cat #14680S 1:1000 |
| Rabbit anti-mono-methyl lysine | Abcam | Cat #ab23366 |
| Alexa-568 Phalloidin | Thermo Fisher Scientific | A12380, 1:40 |
| Alexa-647 Phalloidin | Invitrogen | Cat #A22287, 1:400 |
| Alexa-647 Anti-Rabbit Secondary | Invitrogen | Cat #A-21245, 1:200 |
| Alexa 488 Anti-Rabbit Secondary | Jackson ImmunoResearch | Cat #711-545-152, 1:100 |
| Alexa-568 Anti-Mouse Secondary | Jackson ImmunoResearch | Cat #715-585-150, 1:100 |
| Alexa 488 Anti-Mouse IgG2a Secondary | Invitrogen | Cat #A-21131, 1:200 |
| Alexa-568 Anti-Mouse IgG1 Secondary | Invitrogen | Cat #A-21124, 1:200 |
| IRDye 680RD anti-mouse Secondary | Li-cor | Cat #926-68072, 1:5000 |
| IRDye 800CW anti-rabbit Secondary | Li-cor | Cat #926-32213, 1:5000 |
| Hoechst 3342 | AnaSpec | Cat #AS-83218, 1:2000 |
| **Oligonucleotides and other sequence-based reagents** *For long lists of oligos or other sequences, please refer to the relevant Table(s) or EV Table(s)* | | |
| pLVX-GFP-TTLL12 Forward | This study | CGAATTCGAGGCCGAGCGGGGTCCC |
| pLVX-GFP-TTLL12 Reverse | This study | GGAATTCCTAGACAAGGCAGGTCAACGTG |
| pGEX-6P1-TTLL12 (aa1-260) Forward | This study | CGAATTCGAGGCCGAGCGGGGTCCC |
| pGEX-6P1-TTLL12 (aa1-260) Reverse | This study | GGAATTCCTAGGCCCAGGGCAGCG |

**Table 1.** (continued)

| Reagent/resource | Reference or source | Identifier or catalog number |
|---|---|---|
| pFastBac HTA-TTLL12 Forward | This study | CGAATTCGAGAGGCCGAGCGGGGTC |
| pFastBac HTA-TTLL12 Reverse | This study | GGAATTCCTAGACAAGGCAGGTCAACGTG |
| MDCK TTLL12 KO gRNA #1 | This study | TGTAGAGAATGTCGG |
| MDCK TTLL12 KO gRNA #2 | This study | ACTGGTTCAGCAGCA |
| RPE1 TTLL12 KO gRNA | Horizon Discovery | Cat #CM-014094-04-0002 CTGTACTGGGCAGCTCACCG |
| **Chemicals, enzymes and other reagents** (*e.g., drugs, peptides, recombinant proteins, dyes etc.*) | | |
| Porcine Brain Tubulin | Cytoskeleton Inc. | Cat #T240 |
| Recombinant Histone H3 | EDM Millipore | Cat #14-494 |
| Paclitaxel (Taxol) | Cytoskeleton Inc. | Cat #TXD01 |
| Tubastatin A | Selleckchem | Cat #S8049 |
| Phusion High-Fidelity DNA Polymerase | New England Biolabs | Cat #M0530S |
| Intercept TBS Blocking Buffer | Li-cor | Cat #927-60001 |
| Lipofectamine 2000 Transfection Reagent | Invitrogen | Cat #11668027 |
| tracrRNA | Horizon Discovery | Cat #U-002005-xx |
| DharmaFECT Duo transfection reagent | Horizon Discovery | Cat #T-2010-xx |
| Edit-R-mKate2-Cas9 nuclease mRNA | Horizon Discovery | Cat # CAS11859 |
| **Software** *Include version where applicable* | | |
| GraphPad Prism 9 | https://www.graphpad.com/ | |
| FIJI | https://imagej.net/ij/index.html | |
| UTrack software | | |
| Adobe Illustrator | Adobe Inc. | |
| **Other** (*Kits, instrumentation, laboratory equipment, lab ware etc. that are critical for the experimental procedure and do not fit in any of the above categories*) | | |
| Methyltransferase Fluorometric Assay Kit | Cayman Chemical | Cat #700150 |
| Microtubule Binding Protein Spin-Down Assay Biochem Kit | Cytoskeleton Inc. | Cat #BK029 |

## Cell culture and generation of lentiviral stable cell lines

MDCK II and RPE1 cells were cultured at 37 °C with 5% $CO_2$ in DMEM with 1% penicillin/streptomycin and 10% FBS. GFP and GFP-TTLL12 stable cell lines used in this study were generated using lentiviral transfection. Lentivirus was collected from HEK293T cells, transferred to MDCK target cells, and allowed to incubate for 24 h before replacing with fresh media. Stable population cell lines were then selected using 1 µg/ml of puromycin. Cell lines were routinely tested for mycoplasma and authenticated through STR testing.

## Generation of MDCK and RPE1 CRISPR KO cell lines

MDCK cells stably expressing Tet-inducible Cas9 (Horizon Discovery Edit-R lentiviral Cas9) were grown in a 12-well dish to ~75% confluency followed by treatment with 1 µg/ml doxycycline for 24 h to induce Cas9 expression. Cells were then transfected with

crRNA:tracrRNA mix using the DharmaFECT Duo transfection reagent as described by the Discovery DharmaFECT Duo protocol, available online. Transfected cells were incubated for 24 h, trypsonized, and plated for individual clones. Individual clones were screened by western blot using an anti-TTLL12 antibody.

To generate the RPE1 TTLL12 knockout line, the same protocol was used, except instead of an inducible Cas9 line, the crRNA:tracrRNA mix was co-transfected with Edit-R-mKate2-Cas9 mRNA. Fluorescence-activated cell sorting was performed at the University of Colorado Cancer Center Flow Cytometry Shared Resource Facility using a MoFlo XDP 100, 24 h after transfection, and mKate2-positive cells were plated as individual clones and screened for TTLL12 expression.

## Immunoprecipitation of TTLL12 and proteomics analysis

To identify TTLL12-interacting proteins, MDCK cells expressing GFP alone or GFP-TTLL12 were lysed in 20 mM HEPES pH 7.4,

150 mM NaCl, 1% Triton X-100, 1 mM PMSF, and proteins were immunoprecipitated using a GST-tagged GFP nanobody that was covalently linked to Affigel 10/15 resin. Beads were washed 5 times using 20 mM HEPES pH 7.4, 150 mM NaCl, and 0.1% Triton X-100. Immunoprecipitated proteins were eluted from the beads using 10 mM Tris pH 7.4, 1% SDS, and 100 μM DTT at 55 °C. Mass spectrometry was performed by the University of Colorado Proteomics Core Facility. Proteomics generated a list of 3759 proteins which can be found in Dataset EV1. We narrowed down the list based on the following criteria: enriched in TTLL12-GFP sample at least sixfold as compared to GFP alone control, and we eliminated mitochondrial proteins or proteins involved in transcription. The enrichment cut-off was picked based on enrichment of TTLL12 in GFP-TTLL12 sample (~13-fold). To make sure that we do not miss potentially important TTLL12 binding partners, for initial analysis we chose to focus on protein that enriched 6-fold (half of TTLL12 enrichment). Dataset EV1 lists all proteins identified in the proteomic analysis regardless of enrichment.

## Protein expression and purification

GST-Rab19 was purified from BL21 Codon Plus *E. coli* as previously described (Jewett et al, 2021). GST-TTLL12 (aa1-260) was cloned into pGEX-6P1 and purified from the BL21 Codon Plus *E. coli* strain. Briefly, cultures were grown at 37 °C to an OD of ~0.6 and induced with 0.25 mM IPTG final concentration overnight at 16 °C. Cells were lysed in 20 mM HEPES pH 7.4, 150 mM NaCl, 0.1% Tween 20, 1 mM PMSF, and 1 mM DTT, by French Press. Post-centrifugation lysates were incubated with Glutathione beads for 2 h and washed with 20 mM HEPES pH 7.4, 300 mM NaCl, and 0.1% Tween 20. After washing, GST-TTLL12 (aa1-260) was eluted off the beads using 25 mM glutathione in 20 mM HEPES pH 7.4 and 150 mM NaCl. Eluted protein was dialyzed into PBS with 1 mM DTT.

6His-TTLL12 was cloned into the pFastBac HTA vector and recombinant bacmid was generated using DH10Bac *E. coli* and following the protocol from the Invitrogen Bac-to-Bac Baculovirus Expression system (10359). Baculovirus and Sf9 cells expressing TTLL12 were generated by the University of Colorado Cell Technologies Core Facility. Cells were resuspended in ice-cold 50 mM Tris pH 7.4, 200 mM NaCl, 20 mM imidazole, 1 mM BME, 1 mM PMSF, and 0.1% Triton-X and lysed by French Press. Post-centrifugation lysates were run over a column of Nickel beads. Beads were washed with 50 mM Tris pH 7.4, 300 mM NaCl, 50 mM imidazole, and 0.1% Triton-X. 6His-TTLL12 was eluted off the beads using 150 mM imidazole in lysis buffer. Eluted protein was dialyzed into PBS with 1 mM DTT.

## Generation and purification of TTLL12 polyclonal antibody

Purified GST-TTLL12 (aa1-260) was sent to Pocono Rabbit Farm and Laboratory, Inc. Rabbit anti-sera was affinity purified with Affigel beads coated with recombinant purified 6His-TTLL12 and eluted with 0.1 M Glycine, pH 2.5. The antibody was validated via western blot and using knockout cell lines.

## In vitro binding assays

### Rab19-binding assay

First, recombinant GST-Rab19 was loaded with either GDP or GMP-PNP. For loading, three steps were followed: (1) add EDTA to 5 mM, incubate for 10 min at room temperature; (2) add either GDP or GMP-PNP (non-hydrolyzable GTP analog) to 10 mM, incubate for 10 min at room temperature; (3) add $MgCl_2$ to 15 mM, incubate 10 min at room temperature. After loading, the following reactions were set up: 10 μg GST + 10 μg 6His-TTLL12, 10 μg GST-Rab19 (GDP) + 10 μg 6His-TTLL12, 10 μg GST-Rab19 (GMP-PNP) + 10 μg 6His-TTLL12 in 20 mM HEPES pH 7.4 with 150 mM NaCl and 1% BSA. Reactions were incubated for 1 h at room temperature while rotating. 75 μl of glutathione beads (50% in PBS) were added and incubated for another 30 min while rotating. Beads were washed five times with 1 ml of buffer containing 20 mM HEPES pH 7.4, 300 mM NaCl, and 0.1% Triton X-100. Protein was eluted with 1× SDS sample loading dye, separated by SDS-PAGE, and analyzed via Coomassie blue staining.

### Tubulin-binding assay

The following reactions were set up: 10 μg porcine brain tubulin alone or 10 μg porcine brain tubulin + 10 μg 6His-TTLL12, in 20 mM HEPES pH 7.4 with 150 mM KCl. Reactions were incubated overnight at 4 °C while rotating. 75 μl of $Ni^{+2}$ beads were added and incubated for another 30 min while rotating. Beads were washed five times with 1 ml of buffer containing 20 mM HEPES pH 7.4, 300 mM KCl, and 0.1% Triton X-100. Protein was eluted with 1× SDS sample loading dye, separated by SDS-PAGE, and analyzed via Coomassie blue staining.

For GST-TTLL12 (aa1–260), the following reactions were set up: 10 μg GST + 10 μg porcine brain tubulin or 10 μg porcine brain tubulin + 10 μg GST-TTLL12 (aa1-260), in 20 mM HEPES pH 7.4 with 150 mM KCl. Reactions were incubated overnight at 4 °C while rotating. 75 μl of glutathione beads were added and incubated for another 30 min while rotating. Beads were washed five times with 1 ml of buffer containing 20 mM HEPES pH 7.4, 300 mM KCl, and 0.1% Triton X-100. Protein was eluted with 1× SDS sample loading dye, separated by SDS-PAGE, and analyzed via western blot.

### Competition binding assay

GST-Rab19 was first loaded with GMP-PNP (as described above). After loading, the following reactions were set up: 10 μg 6His-TTLL12 + 10 μg untagged porcine brain tubulin + 0, 5, 10, 20, or 40 μg GST-Rab19 in 20 mM HEPES pH 7.4 with 150 mM KCl. Reactions were incubated overnight at 4 °C while rotating. Overall, 75 μl of $Ni^{+2}$ beads were added and incubated for another 30 min while rotating. Beads were washed five times with 1 ml of buffer containing 20 mM HEPES pH 7.4, 300 mM KCl, and 0.1% Triton X-100. Protein was eluted with 1× SDS sample loading dye, separated by SDS-PAGE, and analyzed via Coomassie blue staining and western blot.

## Microtubule pelleting assay

Protocol was followed as described by the Cytoskeleton Microtubule Binding Protein Spin-Down Assay Biochem Kit. Microtubules were assembled at 35 °C and stabilized with 20 μM of Taxol

final. In total, 5 µg of 6His-TTLL12 was used as the "test" protein. Microtubule pelleting was performed using a Beckman Ultra Centrifuge at 100,000×g. Coomassie blue staining was the detection method of choice.

## Cell lysis and western blot

Unless otherwise stated, cells were lysed on ice in PBS with 1% Triton X-100 and 1 mM PMSF. After 45 min, lysates were clarified at 15,000×g in a prechilled microcentrifuge. Supernatants were collected and analyzed via Bradford assay. In total, 2.5 µg/µl lysate samples were prepared in 4x SDS loading dye, boiled for 5 min at 95 °C, and 30 µg was loaded onto and separated via SDS-PAGE. Gels were transferred onto a 0.45-µm polyvinylidene difluoride membrane, followed by blocking for 30 min in Intercept Blocking Buffer diluted in TBST 1:3. Primary antibodies (diluted in blocking buffer) were incubated overnight at 4 °C. The next day, blots were washed in TBST followed by incubation with IRDye fluorescent secondary antibodies for 30 min at room temperature. Blots were washed again with TBST before imaging on a Li-Cor Odyssey CLx. Densitometry was measured using FIJI.

## Immunofluorescence and confocal microscopy

For RPE1 ciliogenesis experiments, cells were plated on rat tail collagen-coated coverslips and grown to ~80% confluency. To induce ciliogenesis, cells were washed once with PBS and then grown in serum-depleted media (DMEM with 0.5% FBS) for 48 h (or for the indicated time), followed by fixation. For polarized MDCK experiments, 50,000 cells were plated on rat tail collagen-coated Transwell filters (Corning 3460) and grown for 8 days to polarize and reach full ciliation capacity (Jewett et al, 2021).

Cells were fixed with 4% paraformaldehyde for 15 min at room temperature. For the TTLL12 localization experiment, cells were permeabilized with 0.25% Triton-X in PBS for 1 min before fixation.

Cells were then blocked for 1 h at room temperature in block buffer (PBS, 0.1% Triton-X, 10% normal donkey serum). Primary antibodies were diluted in block buffer and incubated overnight at room temperature. Cells were washed with PBST before adding secondary antibodies for 1 h at room temperature. Cells were washed again before mounting in VectaShield and sealing with nail polish. Coverslips for all experiments were no.-1 thickness. All images were acquired using a Nikon Eclipse Ti2 inverted A1 confocal microscope with a 63× oil objective and a z-step size of 0.5 µm. Images in Figs. 4G and EV3A,B,G are widefield images acquired on an inverted Zeiss Axiovert 200 M microscope using a 63× oil objective, QE charge-coupled device camera (Sensicam), and Slidebook v. 6.0 software (Intelligent Imaging Innovations).

## Scratch wound-healing assay

In total, 50 µL of collagen was added to the wells of an imagelock 96-well plate (Sartorius). Collagen was removed and the plate was treated under UV light for 30 min. Extra collagen after UV treatment was aspirated away. RPE cells were resuspended in serum starvation media and counted with the Countess-3 hemocytometer. Cells were diluted to 400,000 cells/mL and 100 µL of cells were added to each well (40,000 cells per well). Six wells were plated for each cell line. The plate was allowed to sit for 15 min, then moved to an incubator for 19 h. Scratches were then made using the Incucyte Woundmaker Tool (Sartorius). Cells were washed twice with serum starvation media and 100 µL of serum starvation media was added to each well. Bubbles were removed with an extra wash when necessary. The plate was placed in an Incucyte S3 system and each well was imaged at ×10 magnification every hour for 25 h. Data were analyzed using the Incucyte Scratch Wound Analysis Software Module. After analysis, wells with improper scratches were discarded. The 1 h timepoint was set as timepoint 0 in order to allow for cell adjustment to the Incucyte incubator. For each technical replicate, data was normalized to the maximum relative wound density value from control cells.

## Microtubule cold stability and nocodazole sensitivity assay

MDCK cells were plated on rat tail collagen-coated coverslips and grown for 24 h. Media was replaced with 4 °C media and the plate was moved to 4 °C for 30 min followed by fixation on ice. For Taxol treatment, cells were treated with 1 µM of Taxol for 10 min before media was replaced with 4 °C media with 1 µM Taxol.

For Nocodazole sensitivity assays MDCK cells were plated on rat tail collagen-coated coverslips and grown for 24 h. Cells were then treated with 500 nM Nocodazole for 10 min, followed by fixation with 4% paraformaldehyde and immunofluorescence analysis using anti-α-tubulin antibodies.

## GFP-MACF18 live-cell imaging and analysis

RPE1 cells were plated on rat tail collagen-coated live-imaging dishes. Cells were transfected with 2 µg GFP-MACF18 using Lipofectamine 2000 in Opti-mem. After 2 h, media was replaced with DMEM, and cells were left overnight. The next day, images were acquired by spinning disk confocal microscopy using a Nikon Ti-E microscope equipped with a 1.45 NA 100x CFI Plan Apo objective, piezoelectric stage (Physick Instrumente), spinning disk confocal scanner unit (CSU10: Yokogawa), 488 nm laser (Agilent Technologies), an EMCCD camera (iXon Ultra 897, Andor Technology), and NIS Elements software (Nikon). Stage was incubated at 37 °C using an ASI 400 Air Stream Incubator (NEVTEK). A z-series covering a 7 µm range at 0.5-µm steps was acquired every 2.5 s for 4 min. GFP-MAC18 comets were analyzed using the UTrack software.

## Scanning electron microscopy

MDCK cells were plated on rat tail collagen-coated Transwell filters and grown for 8 days to polarize and reach full ciliation capacity. Cells were fixed with 2.5% glutaraldehyde in 0.1 M sodium cacodylate buffer overnight at room temperature. Cells were rinsed twice in buffer for 10 min each, then treated with 1% $OsO_4$ for 1 h. Cells were rinsed twice in buffer for 10 min each, followed by dehydration via a graded EtOH series (50%, 70%, 90%, 100%—twice) for 10 min each. Critical point drying ~3 h; Leica CPD300. Sputter coating; 45 s; Leica ACE200. SEM imaging ~30 min per sample on JEOL JSM-6010LA operated at 15 kV by Dr. Eric Wartchow at Colorado Children's Hospital.

## Fluorometric methyltransferase assay

Protocol was followed as described by the Cayman Chemical Methyltransferase Assay Kit manual. For each experiment, a resurfin standard curve was used in duplicate, and samples were assayed in duplicate. The final volume in all wells was 120 µl. All reagents were kept on ice before beginning the assay. In total, 5 µl of 1.5 µg/µl TTLL12 was used as the enzyme "sample" and 10–15 µl of 5 µg/µl porcine brain tubulin was used as an "acceptor". Methyltransferase activity was calculated based on the resurfin standard curve and is per 1 nM TTLL12.

## Image analysis

### MDCK cold stability and Nocodazole sensitivity

Tubulin intensity was measured in FIJI on a maximum projection using an ROI of the same size for every image with the background subtracted. In total, 5–10 cells were imaged per experiment for each condition and the experiment was performed three times.

### TTLL12 localization

TTLL12 intensity was measured using an ROI of the same size for every image at the base of the primary cilium (as indicated by GT335 staining) with the local background subtracted. Four fields of view with ~10 cells each were measured for both WT and TTLL12 KO cells.

### RPE1 ciliation

Ciliation was quantified by counting the number of cells with an Arl13b and GT335 positive cilium divided by the total number of cells in the field of view (indicated by DNA staining). Ten fields of view (50–100 cells per field of view) were averaged per condition per experiment. Cilium length was measured in FIJI using the Arl13b staining.

### MDCK ciliation

Ciliation was quantified by counting the number of cells with the indicated marker (Ac. Tub or Arl13b) co-localizing with gamma-tubulin at the apical surface divided by the total number of cells in the field of view (indicated by DNA staining). Two fields of view (400–600 cells per field of view) were averaged per condition in each experiment. For GT335 staining, ciliation was quantified by the co-localization of GT335 with Arl13b. Arl13b intensity was measured in FIJI on a maximum projection using an ROI of the same size for every measurement. Each measurement was divided by background.

### Spindle positioning analysis in MDCK cells

MDCK WT and MDCK TTLL12 KO cells were grown to confluency and were stained using anti-α-tubulin. To measure mitotic spindle centering the line was drawn through both centrosomes, and the distances from both centrosomes to the plasma membrane were measured (Fig. 7B). The centering index was then calculated by dividing the shorter distance by the longer distance. In all, 10–15 randomly chosen cells were analyzed.

To measure the mitotic spindle angle, two lines were drawn: first one through both centrosomes and second one through major axis of the cell. The angle between those two lines was then measured (Fig. 7C). In total, 5–15 cells were imaged per experiment for each condition and the experiment was performed three times.

## Statistical analysis

All statistical analyses were performed using GraphPad Prism Software. Unless described otherwise, statistics were performed using an unpaired Student's $t$ test (two-tailed) or a one-way ANOVA with a post hoc test assuming normal Gaussian distribution. In all cases, except for TTLL12 localization, data were collected from at least three independent experiments. In imaging experiments where many cells were analyzed (denoted by $n$ in figure legends), statistics were performed on the means from each experiment (denoted by $N$ in figure legends).

# Data availability

No data were deposited in a public database.

# Peer review information

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

## Acknowledgements

We are grateful to Drs. Chad Pearson, Carsten Janke, and Cheryl Walker for very insightful discussions about this project. We would like to thank the University of Colorado Cancer Center Mass Spectrometry Proteomics Shared Resource, Flow Cytometry Shared Resource, and Cell Technologies Shared Resource for their assistance with the mass spec/proteomic analyses, flow sorting, and baculovirus generation, respectively. This work was supported by NIDDK grant R01-DK064380 to RP, as well as T32-GM136444 grant to LC and AJN and Bolie Scholar Award to JC.

## Author contributions

**Julia Ceglowski**: Funding acquisition; Validation; Methodology; Writing—original draft. **Huxley K Hoffman**: Investigation; Writing—review and editing. **Andrew J Neumann**: Investigation; Writing—review and editing. **Katie J Hoff**: Investigation. **Bailey L McCurdy**: Investigation. **Jeffrey K Moore**: Writing—original draft; Writing—review and editing. **Rytis Prekeris**: Conceptualization; Data curation; Supervision; Funding acquisition; Validation; Investigation; Methodology; Writing—original draft; Project administration; Writing—review and editing.

## Disclosure and competing interests statement

The authors declare no competing interests.

# Expanded View Figures

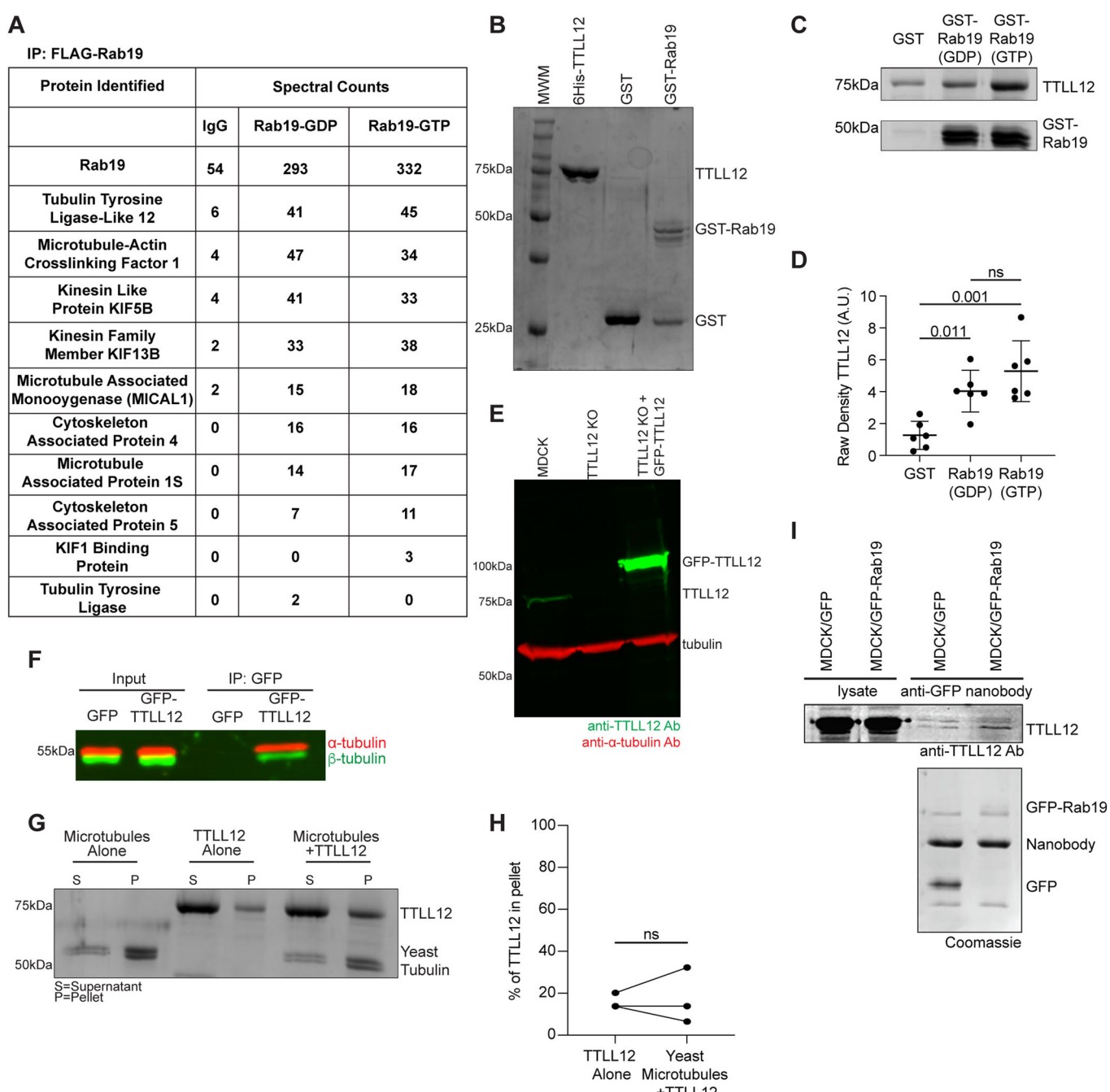

**Figure EV1.** (A) Spectral counts of microtubule-related candidate proteins identified through co-immunoprecipitation/mass spectrometry on FLAG-Rab19 from Jewett et al (2021). (B) Coomassie blue-stained gel with recombinant purified 6His-TTLL12, GST, and GST-Rab19. (C) Binding assay with recombinant GST-Rab19, locked with either GDP or GMP-PNP (GTP), and recombinant 6His-TTLL12 followed by Coomassie blue staining. (D) Quantification of TTLL12 band intensity in (B). Graph shows mean ± SD derived from three independent experiments. Student's *t* test (two-tailed) was used for statistical analysis. (E) Western blot of MDCK WT, TTLL12 KO, and TTLL12 KO cells stably expressing GFP-TTLL12 for TTLL12 (green band). (F) Immunoprecipitation of GPF-TTLL12 followed by western blot for α- and β-tubulin. (G) Microtubule co-precipitation assay. Taxol-stabilized yeast microtubules were mixed with recombinant 6His-TTLL12 and pelleted by centrifugation. The supernatant and pellet were separated and run on a gel followed by Coomassie blue staining. (H) Quantification of TTLL12 in the pellet with or without microtubules from (G). Graph shows mean ± SD from three independent experiments. Student's *t* test (two-tailed) was used for statistical analysis. (I) Immunoprecipitation of GPF-Rab19 or GFP-only from MDCK cells followed by western blot for TTLL12.

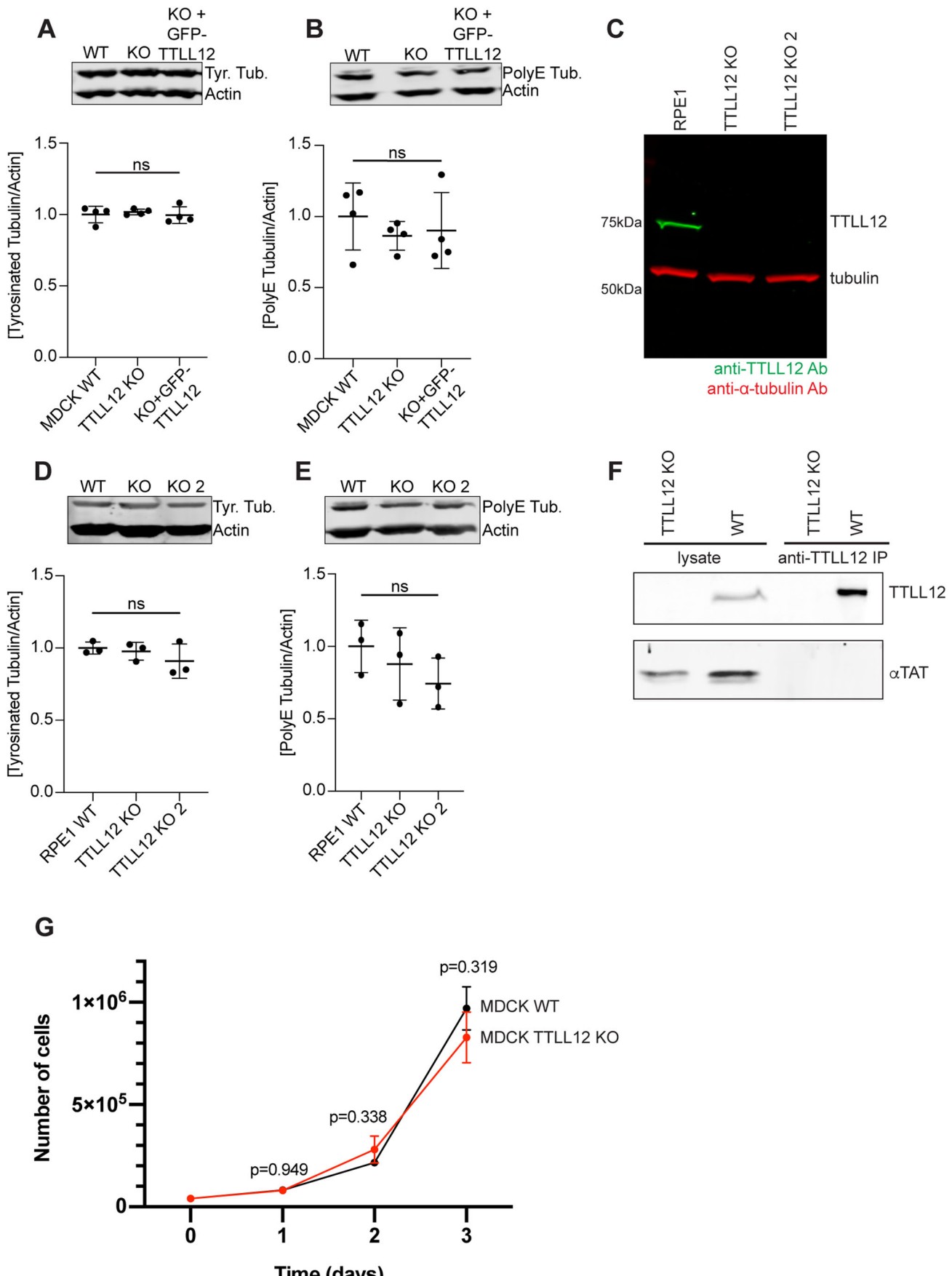

◄  **Figure EV2.** (**A**) Representative western blot and quantification of tyrosinated α-tubulin in MDCK WT, TTLL12 KO, and TTLL12 KO + GFP-TTLL12 cells. Graph shows mean ± SD derived from four independent experiments. Student's *t* test (two-tailed) was used for statistical analysis. (**B**) Representative western blot and quantification of polyglutamylated tubulin in MDCK WT, TTLL12 KO, and TTLL12 KO + GFP-TTLL12 cells. Graph shows mean ± SD derived from four independent experiments. Student's *t* test (two-tailed) was used for statistical analysis. (**C**) Western blot of RPE1 WT, TTLL12 KO, and TTLL12 KO 2 cells for TTLL12 (green band). (**D**) Representative western blot and quantification of tyrosinated α-tubulin in RPE1 WT, TTLL12 KO, and TTLL12 KO 2 cells. Graph shows mean ± SD derived from three independent experiments. Student's *t* test (two-tailed) was used for statistical analysis. (**E**) Representative western blot and quantification of polyglutamylated tubulin in RPE1 WT, TTLL12 KO, and TTLL12 KO 2 cells. Graph shows mean ± SD derived from three independent experiments. Student's *t* test (two-tailed) was used for statistical analysis. (**F**) Western blot of TTLL12 precipitate from wild-type or TTLL12 KO RPE1 cells. (**G**) Proliferation analysis of MDCK WT and MDCK TTLL12 KO cells. One-way ANOVA was used for statistical analysis. The data shown are the means and standard deviations derived from three independent experiments.

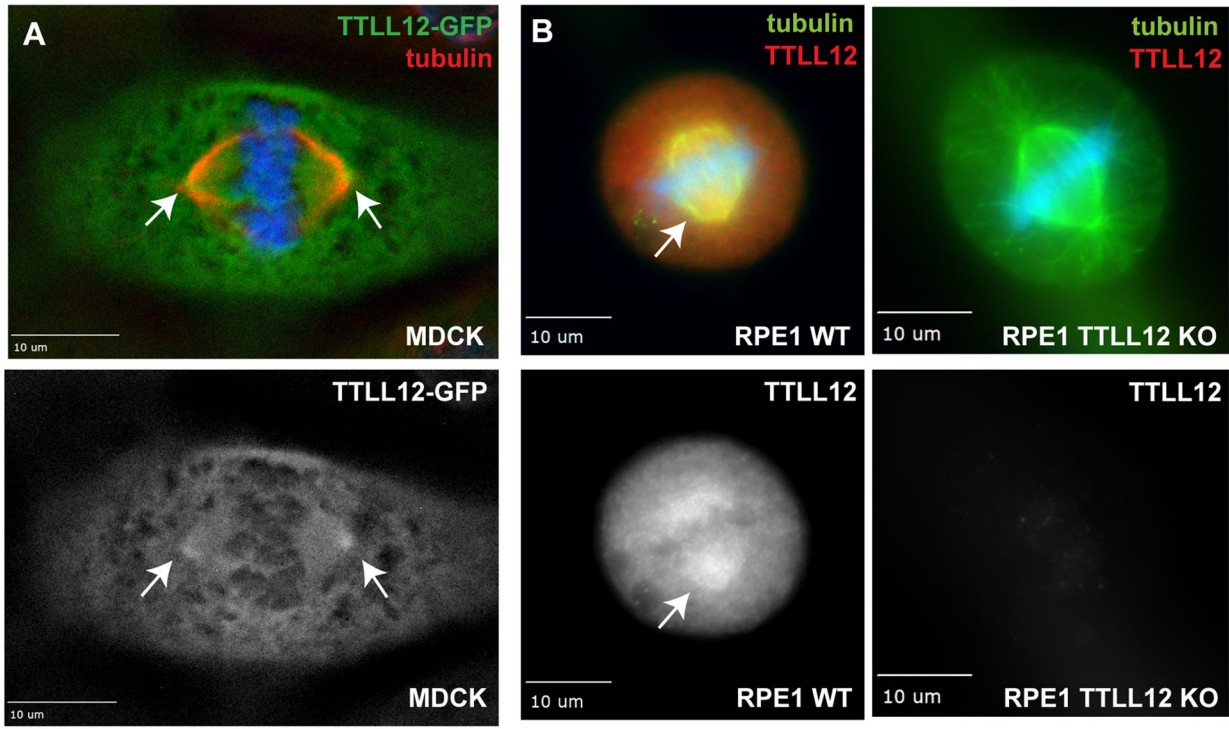

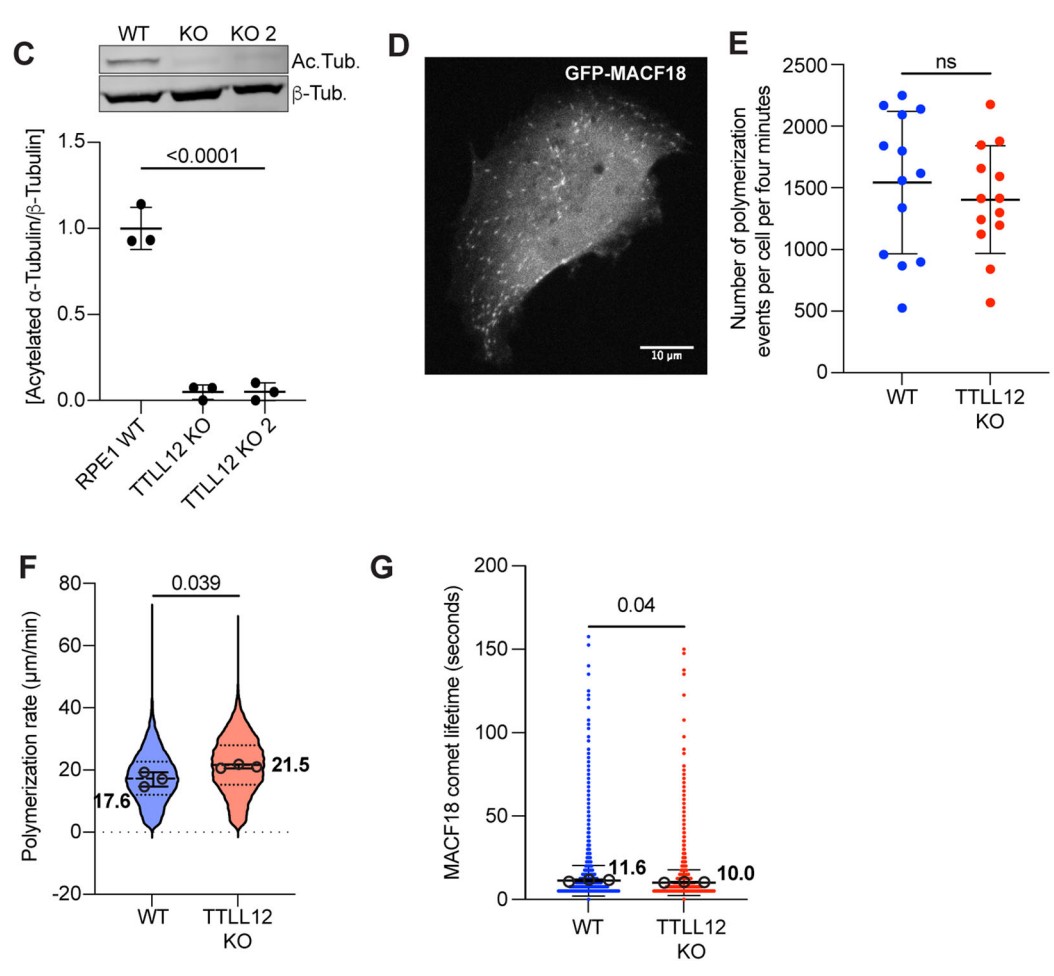

◀ **Figure EV3.** **(A)** MDCK cells stably expressing TTLL12-GFP were stained with anti-α-tubulin antibodies. Arrows point to the mitotic spindle. **(B)** WT or TTLL12 KO RPE cells were stained with anti-TTLL12 (green) and anti-α-tubulin (red) antibodies. Arrow points to the mitotic spindle. **(C)** Representative western blot and quantification of acetylated α-tubulin in RPE1 WT, TTLL12 KO, and TTLL12 KO 2 cells. Graph shows mean ± SD derived from three independent experiments. Student's *t* test (two-tailed) was used for statistical analysis. **(D)** Example image of WT RPE1 cell expressing GFP-MACF18 used for live imaging of microtubule polymerization. Scale bar = 5 µm. **(E)** Quantification of the number of microtubule polymerization events that occur in each cell over the course of 4 min. Images were obtained from three independent experiments. $n = 13$ cells for both WT and TTLL12 KO. Graph shows mean ± SD derived from three independent experiments. Student's *t* test (two-tailed) was used for statistical analysis. **(F)** Quantification of microtubule polymerization rates measured from GFP-MACF18 comets. Violin plot represents all microtubules measured ($n = 20{,}427$ for WT, $n = 19{,}525$ for KO). Black circles represent the average polymerization rate from each independent experiment. Graph shows mean ± SD derived from three independent experiments. Student's *t* test (two-tailed) was used for statistical analysis. **(G)** Quantification of MACF18 comet lifetime. Plot represents all microtubules measured ($n = 20{,}427$ for WT, $n = 19{,}525$ for KO). Black circles represent the average polymerization rate from each independent experiment ($N = 3$). *T* test was performed on the means from the experiment. Student's *t* test (two-tailed) was used for statistical analysis.

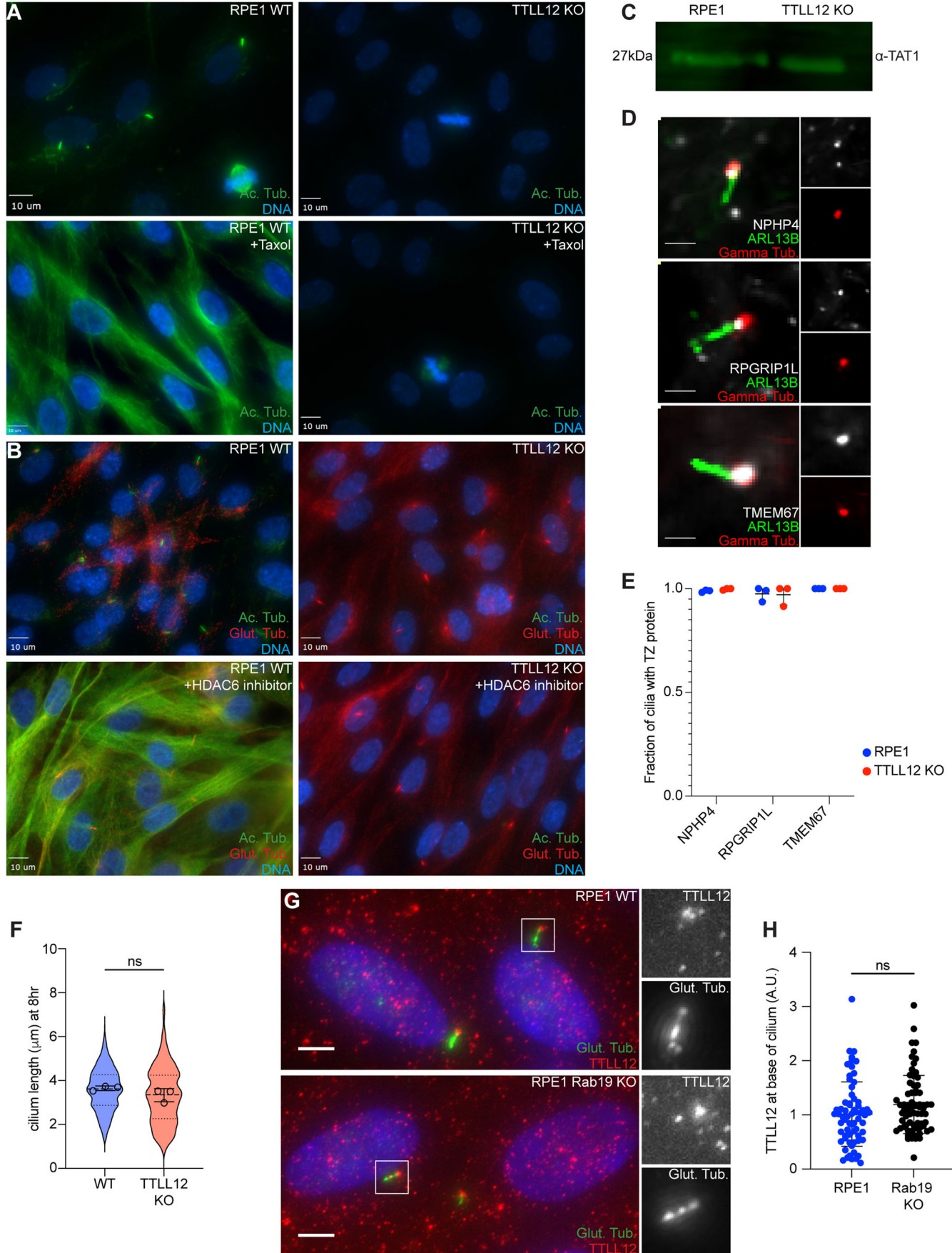

◀  **Figure EV4.** (A) Representative images of RPE1 WT and TTLL12 KO cells with or without Taxol treatment stained for acetylated tubulin and DNA ($N = 1$). (B) Representative images of RPE1 WT and TTLL12 KO cells with or without 1 μm of the pharmacological HDAC6 inhibitor, tubastatin A, and stained for acetylated tubulin, glutamylated tubulin, and DNA ($N = 1$). (C) Western blot of αTAT1 in RPE1 WT and TTLL12 KO cells. (D) Representative images of transition zone proteins NPHP4, RPGRIP1L, and TMEM67 localized to the basal body (gamma tub.) in TTLL12 KO cells. Scale bars: 5 μm. (E) Quantification of RPE1 WT and TTLL12 KO cilia with the respective transition zone proteins from (G). Graph shows mean ± SD derived from three independent experiments. (F) Quantification of primary cilium length after 8 h of serum starvation. Violin plot represents all primary cilia measured. Black circles represent average cilium length ($n = 538$ for WT, $n = 250$ for KO). Shown are the means and standard deviations derived from three independent experiments. Student's *t* test (two-tailed) was used for statistical analysis. (G) Representative images of TTLL12 localization in ciliated RPE1 WT and Rab19 KO cells. Scale bar $= 5$ μm. (H) Quantification of TTLL12 at the base of the primary cilium from G. $N = 2$ and 35 cells were measured in each condition per experiment. Graph shows mean ± SD. Student's *t* test (two-tailed) was used for statistical analysis.

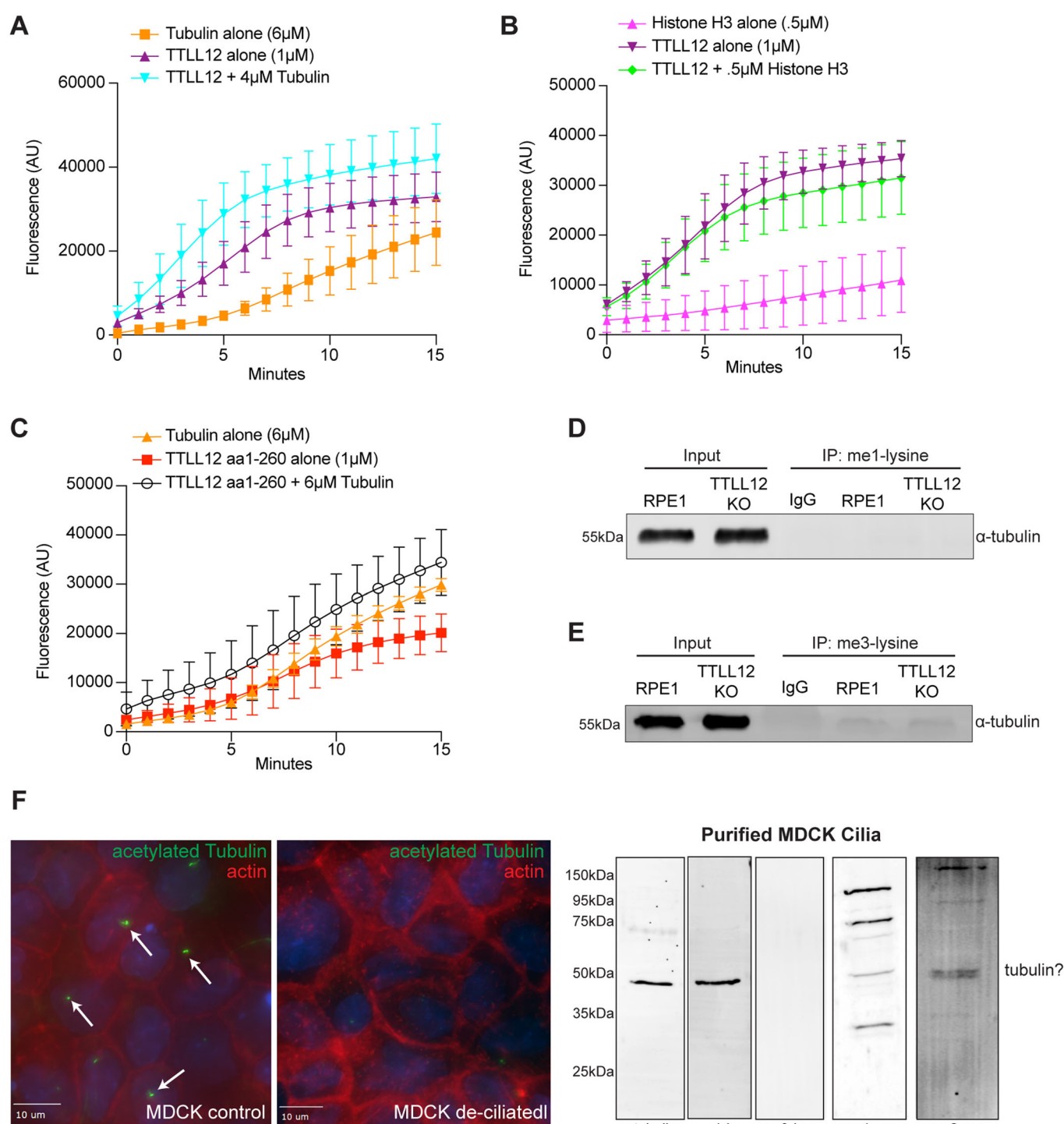

**Figure EV5.** (A) Fluorescence-based assay to measure methyltransferase activity. 1 µM 6His-TTLL12 was incubated with methyl donor SAM and 4 µM porcine brain tubulin and fluorescence was measured over time. Graph shows mean ± SD derived from three independent experiments. (B) 1 µM 6His-TTLL12 was incubated with methyl donor SAM and 0.5 µM histone H3 and fluorescence was measured over time. Graph shows mean and variability derived from two independent experiments. (C) 1 µM GST-TTLL12 aa1-260 was incubated with methyl donor SAM and 6 µM porcine brain tubulin and fluorescence was measured over time. Graph shows mean and variability derived from two independent experiments. (D) Immunoprecipitation of mono-methyl lysine from RPE1 cells followed by western blot for α-tubulin. Left column shows lysates (input) probed for α-tubulin. Right columns show immunoprecipitates probed for α-tubulin. (E) Immunoprecipitation of tri-methyl lysine from RPE1 cells followed by western blot for α-tubulin. Left column shows lysates (input) probed for α-tubulin. Right columns show immune-precipitates probed for α-tubulin. (F) Representative image of MDCK cells incubated in the presence or absence of high calcium deciliation buffer (images on the left). Arrows point to individual cilia. Blots on right are the purified cilia preparations immunoblotted with anti-α-tubulin, anti-me1K, anti-me3K, anti-me1R, and anti-me2R antibodies.

