## [Peer Review File · EMBO Reports]

TTL12 is required for primary ciliary axoneme formation in polarized epithelial cells

Julia Ceglowski, Huxley Hoffman, Andrew Neumann, Katie Hoff, Bailey McCurdy, Jeffrey Moore, and Rytis Prekeris
DOI: 10.15252/embr.202357142

Corresponding author(s): Rytis Prekeris (Rytis.Prekeris@ucdenver.edu)

Review Timeline:

Submission Date:	8th Mar 23
Editorial Decision:	8th May 23
Revision Received:	12th Sep 23
Editorial Decision:	24th Oct 23
Revision Received:	31st Oct 23
Accepted:	9th Nov 23

Editor: Ioannis Papaioannou / Deniz Senyilmaz Tiebe

Transaction Report:

Dear Prof. Prekeris,

Thank you for transferring your research manuscript for consideration by EMBO reports. I apologize for the delayed response, but your manuscript has now been seen by three experts in the field, and we have received the full set of their reports, which are included below.

As you will see, all referees acknowledge that the findings are potentially interesting. Referee #1 is very positive and supports publication of the manuscript in EMBO reports after a few minor corrections. Referees #2 and #3, on the other hand, identify a number of limitations and raise several technical and conceptual concerns, which should be addressed. They also provide many detailed suggestions for the improvement of the study and the manuscript, which should be taken into consideration during revision.

Given these constructive comments, we would like to invite you to revise your manuscript with the understanding that the referee concerns (as detailed in their reports) must be fully addressed and their suggestions taken on board. Please address all referee concerns in a complete point-by-point response. Acceptance of the manuscript will depend on a positive outcome of a second round of review. It is EMBO reports policy to allow a single round of revision only and acceptance or rejection of the manuscript will therefore depend on the completeness of your responses included in the next, final version of the manuscript. If you have any questions or comments, we can also discuss the revisions in a video chat, if you like.

We realize that it is difficult to revise to a specific deadline. In the interest of protecting the conceptual advance provided by the work, we usually recommend a revision within 3 months (August 7th). Please discuss with me the revision progress ahead of this time if you require more time to complete the revisions.

IMPORTANT NOTE:

We perform an initial quality control of all revised manuscripts before re-review. Your manuscript will FAIL this control and the handling will be DELAYED if the following APPLIES:

- 1) If a data availability section providing access to data deposited in public databases is missing.
- 2) If your manuscript contains statistics and error bars based on $n=2$. Please use scatter plots in these cases. No statistics should be calculated if $n=2$.

- 1) A .docx formatted version of the manuscript text (including legends for main figures, EV figures and tables). Please make sure that the changes are highlighted to be clearly visible.
- 2) Individual production quality figure files as .eps, .tif, .jpg (one file per figure). Please download our Figure Preparation Guidelines (figure preparation pdf) from our Author Guidelines pages <https://www.embopress.org/page/journal/14693178/authorguide> for more info on how to prepare your figures.
- 3) A .docx formatted letter INCLUDING the reviewers' reports and your detailed point-by-point responses to their comments. As part of the EMBO Press transparent editorial process, the point-by-point response is part of the Review Process File (RPF), which will be published alongside your paper unless you opt out of this (please see below for further information).
- 4) A complete author checklist, which you can download from our author guidelines (<<https://www.embopress.org/page/journal/14693178/authorguide>>). Please insert information in the checklist that is also reflected in the manuscript. The completed author checklist will also be part of the RPF (please see below for more information).
- 5) Please note that all corresponding authors are required to supply an ORCID ID for their name upon submission of a revised manuscript (<<https://orcid.org/>>). Please find instructions on how to link your ORCID ID to your account in our manuscript tracking system in our Author guidelines (<<https://www.embopress.org/page/journal/14693178/authorguide#authorshipguidelines>>)
- 6) We replaced Supplementary Information with Expanded View (EV) Figures and Tables that are collapsible/expandable online.

A maximum of 5 EV Figures can be typeset. EV Figures should be cited as "Figure EV1, Figure EV2" etc... in the text and their respective legends should be included in the main text after the legends of regular figures.

7) Before submitting your revision, primary datasets (and computer code, where appropriate) produced in this study need to be deposited in appropriate public databases (see <

<https://www.embopress.org/page/journal/14693178/authorguide#dataavailability>>).

Specifically, we would kindly ask you to provide public access to the following datasets/data:

- Mass spectrometry data.

The accession numbers and database should be listed in a formal "Data availability " section (placed after Materials & Methods) that follows the model below (see also < <https://www.embopress.org/page/journal/14693178/authorguide#dataavailability>>):

Data availability

- RNA-seq data: Gene Expression Omnibus GSE46843 (<https://www.ncbi.nlm.nih.gov/geo/query/acc.cgi?acc=GSE46843>)
- [data type]: [name of the resource] [accession number/identifier/doi] ([URL or identifiers.org/DATABASE:ACCESSION])

*** Note: all links should resolve to a page where the data can be accessed. ***

*** Note: the Data Availability Section is restricted to new primary data that are part of this study. ***

8) We request authors to consider both actual and perceived competing interests. Please review the new policy (<<https://www.embopress.org/competing-interests>>) and update your competing interests statement if necessary. Please name this section 'Disclosure and competing interests statement' and place it after the Acknowledgements section.

9) Figure legends and data quantification:

- the name of the statistical test used to generate error bars and P values,
- the number (n) of independent experiments (please specify technical or biological replicates) underlying each data point,
- the nature of the bars and error bars (s.d., s.e.m.)
- If the data are obtained from n {less than or equal to} 2, use scatter plots showing the individual data points.

Discussion of statistical methodology can be reported in the Materials and Methods section, but figure legends should contain a basic description of n, P and the test applied.

10) We now request publication of original source data with the aim of making primary data more accessible and transparent to the reader. Our source data coordinator will contact you to discuss which figure panels we would need source data for and will also provide you with helpful tips on how to upload and organize the files.

11) Our journal encourages inclusion of *data citations in the reference list* to directly cite datasets that were re-used and obtained from public databases. Data citations in the article text are distinct from normal bibliographical citations and should directly link to the database records from which the data can be accessed. In the main text, data citations are formatted as follows: "Data ref: Smith et al, 2001" or "Data ref: NCBI Sequence Read Archive PRJNA342805, 2017". In the Reference list, data citations must be labeled with "[DATASET]". A data reference must provide the database name, accession number/identifiers and a resolvable link to the landing page from which the data can be accessed at the end of the reference.

Further instructions are available at <<https://www.embopress.org/page/journal/14693178/authorguide#referencesformat>>.

12) Please also note our reference format:

<<http://www.embopress.org/page/journal/14693178/authorguide#referencesformat>>.

13) We now use CRediT to specify the contributions of each author in the journal submission system. CRediT replaces the author contribution section, which should be removed from the manuscript. Please use the free text box to provide more detailed descriptions. See also guide to authors:

<<https://www.embopress.org/page/journal/14693178/authorguide#authorshipguidelines>>.

14) As part of the EMBO publications' Transparent Editorial Process, EMBO reports publishes online a Review Process File to accompany accepted manuscripts. This File will be published in conjunction with your paper and will include the referee reports, your point-by-point response and all pertinent correspondence relating to the manuscript.

You can opt out of this by letting the editorial office know (emboreports@embo.org). If you do opt out, the Review Process File link will point to the following statement: "No Review Process File is available with this article, as the authors have chosen not to make the review process public in this case."

I look forward to seeing a revised version of your manuscript when it is ready. Please let me know if you have any questions or comments regarding the revision.

Yours sincerely,

Ioannis Papaioannou, PhD
Editor
EMBO reports

Referee #1:

Hoff and others identified the Tubulin Tyrosine Ligase-Like (TTLL) family protein number 12 (TTLL12) localizes to the base of the primary cilium and is required for cilia formation in polarized renal epithelial cells. They further showed that TTLL12 is directly bound to the $\alpha\beta$ -tubulin heterodimer in vitro and involved in regulating microtubule dynamics, stability, and post-translational modifications (PTMs). For PTMs, TTLL12 differs from other known TTLL proteins that catalyze the addition of glutamate or glycine to microtubule C-terminal tails in that it promotes microtubule lysine acetylation and arginine methylation as well. They finally conclude that they identified a novel microtubule regulator and provides insight into the requirements for apical extracellular axoneme formation.

The authors showed us strong evidence that TTLL12 is required for ciliary formation of polarized epithelial cells. The molecular mechanism underlying this event remains partially elusive though many possibilities were given by providing very interesting observations. Other than these, the authors have clearly described their findings and showed strong biochemical evidence to support their conclusion/prediction. Therefore, I have no much comments for their scientific merits but do ask them to correct a few errors that was obviously existing.

- 1) In the reference list, please remove the number in page 14 and reformat the list.
- 2) Figure 3A needs to be replaced with one of better quality.

Referee #2:

Understanding the role of tubulin posttranslational modifications (PTMs) in regulating microtubule functions has been limiting. And when it comes to their role in regulating mammalian primary cilia, it is almost an enigma barring a couple of studies, with

most of the studies pertaining to tubulin acetylation and glutamylation. The predominant family of tubulin PTM enzymes belong to the family of Tubulin Tyrosine Ligase-like (TTL) proteins and while the role of most of the enzymes in this family has been well-established, not much was known about the role of TTL12.

In their current manuscript, Ceglowski et al have identified TTL12 as an interactor of Rab19, the small GTPase which is a critical factor of ciliogenesis. The authors go on to establish that unlike other members of the TTL family, TTL12 binds to the α/β heterodimer and not microtubules of the ciliary axoneme. They also establish that TTL12 plays a role in stabilizing the microtubules and influence their dynamics. Interestingly, the authors find that loss of TTL12 leads to reduction in microtubule acetylation, a process catalysed by the α TAT-1 enzyme, and one which occurs in the lumen of the microtubules. They then go on to establish that this loss of acetylation does not impact ciliogenesis in non-polarized cells that have intracellular pathway of ciliation, but abrogate cilia formation in polarized cells that have an extracellular ciliogenesis mechanism. Finally, they provide a novel function for an enzyme belonging to the TTL family in that they show that TTL12 is involved in α -tubulin arginine-methylation, which has not been reported till date.

Overall, the manuscript provides a novel, fundamental insight into the role of TTL12 with an implication for its role in axoneme biogenesis, dynamics, stability and in regulating specific tubulin PTMs that are divergent from those that other enzymes of the TTL family catalyse. The manuscript is well-written and the data is very succinctly outlined. However, there are some concerns that need to be addressed before the manuscript can be considered for publication in the journal.

-
Major points:

1. How do Rab19 and TTL12 interact? Since the authors show that Rab19 did not inhibit the binding of TTL12 to tubulin, do they have any speculation or data to support how Rab19 and TTL12 are interacting? Is it a direct interaction or is the interaction facilitated by the tubulin itself?
2. Since the authors show the interaction between Rab19, TTL12 and tubulin in vitro, this does not give the conclusion that the same interaction occurs within the cilia as well? The authors do not give any cellular interaction details in the manuscript. The authors need to show some evidence to say in cellulo there is a direct interaction between the two proteins.
3. In figure 3A, the authors claim that there is a statistically significant difference in the levels of acetylation of tubulin in MDCK cells KO for TTL12. However, the blot image represented fails to highlight this aspect. I would like to see a better representative blot of the reduction of tubulin acetylation to be able to support the authors claim that TTL12 KO affects tubulin acetylation.
4. TTL12 binds to the tubulin dimer influencing acetylation, but acetylation of microtubules by α TAT1 happens in the microtubules. The authors discuss the possibilities of how this whole process is regulated, without looking at possible interaction between the two enzymes. While it is understandable that it requires substantial work to delineate the process, as a start, it would be interesting to see if TTL12 interacts with α TAT1 and whether this happens on microtubules. If this is analyzed by the authors by simple immunofluorescence staining for TTL12 and α TAT1, or by Co-IP analyses in their cells, it would already give an initial insight into the process, strengthening the message of the current manuscript.
5. In figure 3B, since the overexpression of GFP-TTL12 TTL12 KO cells did not rescue loss of acetylation, it appears that the protein is not active. Did the authors test the activity of GFP-TTL12 to ensure that the protein is active? Since the authors also made the TTL12 KO Rpe-1 cells, which showed almost complete loss of tubulin acetylation, the authors can use this cell line to test the activity of their GFP-TTL12 by transient transfection and show whether this rescues the phenotype or not.
6. While The MACF18 comet-based assay is a good readout, the authors' claim that in the absence of TTL12, the comet lifetime is reduced is unconvincing. Looking at their quantification, a reduction in the mean value from 11.6 s in wildtype to 10 s in TTL12 KO cannot be considered significant. It would be good if the authors rescue this effect by overexpressing TTL12, to show that the growth speed of microtubules slows down and more comets have a longer lifespan.
7. How does TTL12 impact the stability of the microtubules and their dynamics in vitro? Have the authors tested the dynamics of growth and catastrophe of the microtubules using an in vitro microtubule polymerization assay?
8. Previous studies have shown that acetylation of tubulin renders them more resistant to depolymerizing agents like nocodazole. Did the authors test if the TTL12 KO cells that totally lack tubulin acetylation if they are more prone to such depolymerization?
9. The authors have mainly focused on the role of TTL12 in regulating axonemal microtubules, while the effect of microtubule dynamics in their TTL12 KO cells is global. What is the effect of loss of TTL12 on general cell physiology? Since previous studies (Hubbert C et al., Nature, 2002; Palazzo et al, Nature. 2003) have shown that reduced acetylation could lead to increased cell motility, it would be important to check this aspect in TTL12 KO cells, to determine if the role of TTL12 is cilia specific or has a global cellular effect.

10. Since polarized MDCK cells lose ciliogenesis upon loss of TTLL12, does this then make these cells more proliferative or do the cells lose their polarity? Can the authors comment on this?

11. The authors have used MDCK cells to assess the effect of TTLL12 on extracellular ciliogenesis and show in Figure 5A that while there is no change in apical cortical actin near the basal bodies, the level of acetylation of axonemal microtubules goes down in the TTLL 12 KO MDCK cells. The authors then go on to state that this reduction in acetylation is rescued by overexpressing TTLL12. The concern here is that while the authors show some GFP positive cells in the KO + GFP-TTLL12 panel in Figure 5A, almost all the cells in that field appear to have cilia with Ac-tubulin staining. And again, there appears to be no effect on the cytosolic microtubules. How do the authors explain this?

12. The authors predominantly focus on the role of TTLL12 in regulating axonemal microtubules since it interacts with Rab19, which is required for proper ciliogenesis. However, when it comes to testing the functional mechanism of how TTLL12 regulates the whole process, they fail to show any evidence that is pertaining to the axonemes. Their studies on arginine methylation represented in Figures 6 and S4 are carried out either using porcine brain tubulin or the total pool of cellular tubulin. Whether methylated tubulin is present in the axonemes or not is something they have totally failed to address. The authors have to clearly demonstrate that the methylated tubulin is incorporated within the cilia, to emphasize that the effect of TTLL12 is majorly on axonemal microtubules.

Minor points:

1. Why do the authors think that TTLL12 is not a canonical Rab19 effector protein? There is no data in the manuscript that supports this claim. The authors could provide some clarity on their statement in the text, or could show some data supporting their claim.

2. The IP-mass spec data shown in table S1 by the authors does not corroborate with their earlier IP done in 2021 (Jewett et al., 2021). If there is a direct interaction between TTLL12 and Rab19, then why do they not see a similar enrichment of Rab19 in their GFP-TTLL12 IP-mass spec?

- Is it due to the interference by GFP?

- Is it due to the differences in the experimental conditions used for the two IPs?

- The reason I ask this is because their data to show direct interaction is an in vitro system, while their IP is using lysates of MDCK cells

expressing TTLL12. I wonder if the interaction is direct even in vivo / in cellulo, similar to what the authors see in vitro.

3. What was the rationale behind choosing specifically TTLL12 from the different microtubule-associated proteins the authors identified in their Rab19 interactome? And what was the rationale behind selecting an enrichment of >6-fold enrichment? The authors need to clarify this in their manuscript.

4. Typographical error: Page 3: "...which exhibits high conservation as compared to mammalian tubulin, but has no tubulin tail PTMs...."

5. Typographical error: Page 3: ".....TTLL12 and the microtubule polymer may not that be...." has to be: may be not that

6. The supplementary moves 1 and 2 provided by the authors does not clearly say which cell is the WT and which cell is the KO. IT is difficult to make out from the video to understand which one has the faster speed of polymerization as suggested by the authors in Figure 3I. The authors can slow down the speed of the video to be able to appreciate this aspect with better clarity.

7. It was not clear to me why the authors analyzed the presence of transition zone in their TTLL12 KO cells, since the cells show normal cilia formation. This is already an indication that the transition zone is normal.

8. The immunofluorescence microscopy image of Rpe1 cells stained with the TTLL12 polyclonal antibody in Figure 4G and Figure S3G are not very convincing. Suggesting that TTLL12 is localizing to the base of the cilia from this image is quite speculative as one of the cells with GT335 positive cilia in Figure 4G does not show any localization of TTLL12 at the base of the cilia. Moreover, from this image it is not conclusive whether the TTLL12 is at the base of the cilium or the tip, as centriolar markers are missing. It would be good to have a better representative image of this, with centriolar marker like gamma tubulin or centrin to highlight the base of the cilium.

9. It is quite difficult to differentiate the signals for Ac-tub and actin in Figure 5A. The authors are recommended to use different colors like blue or white to denote one of these proteins for better visualization and understanding of the observations.

10. Typographical error: Page 9: ".....depleting TTLL12 in cells does not increase affect tyrosination...." It should be does not affect.

11. Typographical error: Page 9: "...While this is suggestive of TTLL12...."

Referee #3:

The Manuscript from Ceglowski et al addresses the function of the TTLL12 protein in cilia formation and general MT PTMs and stability. TTLL12 was identified as a binding partner for Rab19 which the authors have previously shown is involved in cilia formation. TTLL12 is a member of a large TTLL family, some of which are known to localize to the basal bodies and affect cilia, yet TTLL12 remained poorly characterized and had no known enzymatic activity. The authors perform a number of biochemical analysis of MT binding, PTMs, and stability as well as imaging analysis of cilia formation in both RPE1 cells and polarized MDCK. They claim that TTLL12 has an important role in regulating MT stability and in regulating cilia formation in exposed cilia of polarized cells but not in the submerged cilia of RPE1 cells. There is a lot of nice data in this paper but I found the story to be a bit incomplete and confusing. While the authors admit and discuss the limitations of direct versus indirect effects of TTLL12 on tubulin PTMs, the end result is a confusing combination of phenotypes where the reader still has little understanding of the function of TTLL12. While the data is interesting, I think there needs to be a little more mechanistic insight before it is up to the level of EMBO Reports.

Comments:

One of the main claims of the paper is that TTLL12 has a functional difference on submerged versus polarized cilia. I think more needs to be done to substantiate this claim. It could be that these two cells have different levels of other TTLLs that compensate or some other cell type difference unrelated to the type of cilia. Mazo et al. showed that cilia submersion was regulated by sub distal appendages in RPE1 cells. Perhaps the best experiment I can think of to properly test the authors model would be to generate RPE1 cells that have an exposed cilia and to see if there is now a defect similar to MDCK. The formation of cilia in RPE1 cells that are not acetylated is intriguing. Are these cilia stable? Are they functional?

The OE MDCK cell line is massively over expressed relative to the endogenous levels. Given that, 2 spectral counts for Rab19 is pretty unimpressive and might argue against a strong interaction. It would be nice to see what other proteins bind TTLL12, this data set seems pretty cherry picked and it would be nice to know what else it interacts with and perhaps with those counts the relevance of Rab19 spectral counts would be more clear.

2b...given the variability in this assay result it seems worthwhile to perform several more iterations. It seems like the one outlier might be affecting an important interpretation.

Are dimers actually acetylated. I feel like there is a real conceptual break in what the authors are proposing. It is simply not clear how a protein that only interacts with non polymer MTs is affecting MT polymers. While the authors discuss this at length without experimental insight we simply don't know what TTLL12 is doing. They have observed a lot of changes to MTs but what is direct and what is indirect are left unanswered.

In MDCK cells the rescue causes a dramatic increase in ciliation, which is likely more of an overexpression than a rescue but is an interesting result that should be discussed.

"We propose that TTLL12 actively functions at the basal body and interacts with the α/β -tubulin heterodimer before incorporation into the primary cilium; this interaction promote microtubule stability and tubulin PTMs required for building an axoneme (Fig. 7)"....Perhaps the most fascinating data presented is the fact that the RPE cells generate a cilium without acetylation, which is pretty amazing, but is also inconsistent with this statement. Additionally, the fact that Arl13b is at the centrosome in MDCKs, suggests that cells are starting to make cilia but that they are just short and underdeveloped rather than unciliated.

We greatly appreciate all constructive comments from reviewers. We have incorporated vast majority of proposed changes and new experiments and believe that all these changes substantially improved the manuscript. For point-by-point changes see list below. All additions to the manuscript text are marked in yellow.

Referee #1:

Hoff and others identified the Tubulin Tyrosine Ligase-Like (TTLL) family protein number 12 (TTLL12) localizes to the base of the primary cilium and is required for cilia formation in polarized renal epithelial cells. They further showed that TTLL12 is directly bound to the α/β -tubulin heterodimer in vitro and involved in regulating microtubule dynamics, stability, and post-translational modifications (PTMs). For PTMs, TTLL12 differs from other known TTLL proteins that catalyze the addition of glutamate or glycine to microtubule C-terminal tails in that it promotes microtubule lysine acetylation and arginine methylation as well. They finally conclude that they identified a novel microtubule regulator and provides insight into the requirements for apical extracellular axoneme formation.

The authors showed us strong evidence that TTLL12 is required for ciliary formation of polarized epithelial cells. The molecular mechanism underlying this event remains partially elusive though many possibilities were given by providing very interesting observations. Other than these, the authors have clearly described their findings and showed strong biochemical evidence to support their conclusion/prediction. Therefore, I have no much comments for their scientific merits but do ask them to correct a few errors that was obviously existing.

1. In the reference list, please remove the number in page 14 and reformat the list.

As suggested, we have reformatted the reference list.

2. Figure 3A needs to be replaced with one of better quality.

As suggested, we replaced Figure 3A image.

Referee #2:

Understanding the role of tubulin posttranslational modifications (PTMs) in regulating microtubule functions has been limiting. And when it comes to their role in regulating mammalian primary cilia, it is almost an enigma barring a couple of studies, with most of the studies pertaining to tubulin acetylation and glutamylation. The predominant family of tubulin PTM enzymes belong to the family of Tubulin Tyrosine Ligase-like (TTLL) proteins and while the role of most of the enzymes in this family has been well-established, not much was known about the role of TTLL12.

In their current manuscript, Ceglowski et al have identified TTLL12 as an interactor of

Rab19, the small GTPase which is a critical factor of ciliogenesis. The authors go on to establish that unlike other members of the TTLL family, TTLL12 binds to the α/β heterodimer and not microtubules of the ciliary axoneme. They also establish that TTLL12 plays a role in stabilizing the microtubules and influence their dynamics. Interestingly, the authors find that loss of TTLL12 leads to reduction in microtubule acetylation, a process catalysed by the α TAT-1 enzyme, and one which occurs in the lumen of the microtubules. They then go on to establish that this loss of acetylation does not impact ciliogenesis in non-polarized cells that have intracellular pathway of ciliation, but abrogate cilia formation in polarized cells that have an extracellular ciliogenesis mechanism. Finally, they provide a novel function for an enzyme belonging to the TTLL family in that they show that TTLL12 is involved in α -tubulin arginine-methylation, which has not been reported till date.

Overall, the manuscript provides a novel, fundamental insight into the role of TTLL12 with an implication for its role in axoneme biogenesis, dynamics, stability and in regulating specific tubulin PTMs that are divergent from those that other enzymes of the TTLL family catalyse. The manuscript is well-written and the data is very succinctly outlined. However, there are some concerns that need to be addressed before the manuscript can be considered for publication in the journal.

Major points:

1. How do Rab19 and TTLL12 interact? Since the authors show that Rab19 did not inhibit the binding of TTLL12 to tubulin, do they have any speculation or data to support how Rab19 and TTLL12 are interacting? Is it a direct interaction or is the interaction facilitated by the tubulin itself?

It is likely that Rab19 and TTLL12 binds directly to each other since we can show this binding using purified 6His-TTLL12 and GST-Rab19 (see Supplemental Figure 1B-D). While we do not know the domain within TTLL12 that mediate binding to Rab19 (is one of our goals in future studies), we speculate that it is likely outside the TTLL-domain, since TTLL12 can co-bind purified Rab19 and purified porcine tubulin. We actually wonder whether Rab19 may be required to stimulate presumptive TTLL12 methyltransferase activity. Future studies, however, will be needed to test that. We added a few sentences to discussion section to discuss this possibility.

2. Since the authors show the interaction between Rab19, TTLL12 and tubulin in vitro, this does not give the conclusion that the same interaction occurs within the cilia as well? The authors do not give any cellular interaction details in the manuscript. The authors need to show some evidence to say in cellulo there is a direct interaction between the two proteins.

We added immunoprecipitation (using anti-GFP nanobody) data from cells expressing either GFP alone or GFP-Rab19 (see supplemental figure 1I). This data also consistent with Rab19 interacting with TTLL12. With this new data we now have 5 lines of

evidence that Rab19 and TTLL12 interact in vivo and in cellulo: (1) Rab19 was identified in GFP-TTLL12 proteomics (see supplemental table 1); (2) TTLL12 was identified in FLAG-Rab19 proteomics (previously published in PMID:33561422); (3) purified GST-Rab19 binds to purified 6His-TTLL12; (4) TTLL12 co-precipitates with GFP-Rab19; (5) both Rab19 and TTLL12 are present at the basal body (basal body localization of TTLL12 shown in this manuscript while basal body localization of Rab19 is shown in paper PMID:33561422).

3. In figure 3A, the authors claim that there is a statistically significant difference in the levels of acetylation of tubulin in MDCK cells KO for TTLL12. However, the blot image represented fails to highlight this aspect. I would like to see a better representative blot of the reduction of tubulin acetylation to be able to support the authors claim that TTLL12 KO affects tubulin acetylation.

As suggested, we replaced Figure 3A image.

4. TTLL12 binds to the tubulin dimer influencing acetylation, but acetylation of microtubules by α TAT1 happens in the microtubules. The authors discuss the possibilities of how this whole process is regulated, without looking at possible interaction between the two enzymes. While it is understandable that it requires substantial work to delineate the process, as a start, it would be interesting to see if TTLL12 interacts with α TAT1 and whether this happens on microtubules. If this is analyzed by the authors by simple immunofluorescence staining for TTLL12 and α TAT1, or by Co-IP analyses in their cells, it would already give an initial insight into the process, strengthening the message of the current manuscript.

As suggested, we tested whether TTLL12 and α TAT1 interact, thus, allowing TTLL12 regulate α TAT1 activity. To that end, we immunoprecipitated TTLL12 and blotted for α TAT1. As shown in supplemental figure 2F, α TAT1 did not co-precipitate with TTLL12. While it does not unequivocally prove that TTLL12 does not bind α TAT1, it is certainly not consistent with this idea. We also tried to do immunofluorescent microscopy to test whether both proteins co-localize. However, α TAT1 is present mostly in cytosol and we did not observe it in basal body, thus, making this analysis not very informative (consequently, we decided not to include IF analysis of α TAT1 into manuscript).

5. In figure 3B, since the overexpression of GFP-TTLL12 TTLL12 KO cells did not rescue loss of acetylation, it appears that the protein is not active. Did the authors test the activity of GFP-TTLL12 to ensure that the protein is active? Since the authors also made the TTLL12 KO Rpe-1 cells, which showed almost complete loss of tubulin acetylation, the authors can use this cell line to test the activity of their GFP-TTLL12 by transient transfection and show whether this rescues the phenotype or not.

Since we can rescue ciliation, we do believe that TTLL12-GFP is likely active. That is also consistent with the new data that we added to the manuscript showing that TTLL12-GFP rescues decrease in microtubule stability (see data Figure 3). One possibility is that decrease in tubulin acetylation could be result of some sort of

compensatory event to decrease in microtubule stability that we still do not fully understand. We are very much interested in figuring this out, but we also feel that these studies are outside the scope of this manuscript.

Using RPE cells to try to rescue TTLL12-KO induced decrease in acetylation would be an excellent idea. However, RPE cells are puromycin resistant (and already express tet-inducible Cas9 using G418 selection), making generation of stably expressing RPE1 cell lines very difficult. Furthermore, RPE1 transient transfection efficiency is also very low, making it difficult to use transient expressions for rescue experiments.

6. While the MACF18 comet-based assay is a good readout, the authors' claim that in the absence of TTLL12, the comet lifetime is reduced is unconvincing. Looking at their quantification, a reduction in the mean value from 11.6 s in wildtype to 10 s in TTLL12 KO cannot be considered significant. It would be good if the authors rescue this effect by overexpressing TTLL12, to show that the growth speed of microtubules slows down and more comets have a longer lifespan.

As mentioned above, the RPE1 cells that we use to generate knock out are already resistant to puromycin (all RPE1 cells are resistant to it) and G418 (our RPE KO cell lines express tet-inducible Cas9). Thus, to do these experiments we would need to transiently express (in the same cell) both, GFP-MACF48 and TTLL12-mCherry, which is not trivial in RPE1 cells. Interestingly, the defects in microtubule stability are much more pronounced in mitotic spindle microtubules (see fig3 and supp fig3) rather than interphase microtubules. Considering that TTLL12 is present at the basal bodies as well as centrosomes of the mitotic spindle (see new data in supplemental figure 3A-B), but not interphase centrosome, we wondered whether measuring interphase microtubule dynamics is not the best approach to test the role of TTLL12. Consequently, in this revised manuscript we moved GFP-MACF18 data into supplemental figure 3. Furthermore, we added new data analyzing the role of TTLL12 in mediating resistance of spindle microtubules to nocodazole (see figure 3). Importantly, since these are MDCK cells, we could use MDCK-TTLL12 KO/TTLL12-GFP cells to show that TTLL12-GFP does rescue TTLL12-KO induced sensitivity of mitotic spindle microtubules to nocodazole.

7. How does TTLL12 impact the stability of the microtubules and their dynamics in vitro? Have the authors tested the dynamics of growth and catastrophe of the microtubules using an in vitro microtubule polymerization assay?

We agree that these would be very interesting assays to do. However, these are not-trivial assays to set up and perform. While we are definitely planning to try that in our future experiments, we feel that this is beyond the scope of this manuscript.

8. Previous studies have shown that acetylation of tubulin renders them more resistant to depolymerizing agents like nocodazole. Did the authors test if the TTLL12 KO cells that totally lack tubulin acetylation if they are more prone to such depolymerization?

As suggested, we performed the experiments testing the role of TTLL12 in regulating mitotic spindle microtubule sensitivity to nocodazole. The new data (see fig 3) is fully consistent with our hypothesis that TTLL12 increases microtubule stability. Importantly, TTLL12 KO effects can be rescued by over-expressing TTLL12-GFP in MDCK cells.

9. The authors have mainly focused on the role of TTLL12 in regulating axonemal microtubules, while the effect of microtubule dynamics in their TTLL12 KO cells is global. What is the effect of loss of TTLL12 on general cell physiology? Since previous studies (Hubbert C et al., Nature, 2002; Palazzo et al, Nature. 2003) have shown that reduced acetylation could lead to increased cell motility, it would be important to check this aspect in TTLL12 KO cells, to determine if the role of TTLL12 is cilia specific or has a global cellular effect.

Our data (including newly added data) suggest that TTLL12 may be important in formation and function of specialized microtubule structures, such as cilia and mitotic spindle. Since previous studies (that reviewer refers to) also implicates acetylated tubulin in cell migration we added (as suggested by reviewer) some new data showing that TTLL12 appears to be present at the leading edge of the migrating cell and that TTLL12 KO increases cell migration. That is fully consistent with previous studies showing that reduced acetylation increases cell migration. Additionally, we also investigated the effect of TTLL12 KO on mitotic spindle. Our data shows that TTLL12 appears to be required for mitotic spindle positioning in MDCK cells. All these data are now presented in new Figure 6.

10. Since polarized MDCK cells lose ciliogenesis upon loss of TTLL12, does this then make these cells more proliferative or do the cells lose their polarity? Can the authors comment on this?

As suggested, we have tested proliferation of MDCK TTLL12 KO cells. It turned out that there is no big change in proliferation, thus, decreased ciliation is not due to increase in proliferation (see supplemental figure 2G).

11. The authors have used MDCK cells to assess the effect of TTLL12 on extracellular ciliogenesis and show in Figure 5A that while there is no change in apical cortical actin near the basal bodies, the level of acetylation of axonemal microtubules goes down in the TTLL 12 KO MDCK cells. The authors then go on to state that this reduction in acetylation is rescued by overexpressing TTLL12. The concern here is that while the authors show some GFP positive cells in the KO + GFP-TTLL12 panel in Figure 5A, almost all the cells in that field appear to have cilia with Ac-tubulin staining. And again, there appears to be no effect on the cytosolic microtubules. How do the authors explain this?

Our ciliation data shows that TTLL12 KO does not completely block cilia formation but rather significantly decreases it (see figure 5A, C and E), thus, some of the cells lacking TTLL12 will still form cilia. We speculate that cells lacking TTLL12 may lose cilia (rather than fail to form it) as the result of decrease in axonemal microtubule stability. We are

planning to test that hypothesis in future studies and added short discussion regarding that to Discussion section.

We totally agree that loss of TTLL12 does not seem to dramatically affect interphase microtubules. That is actually consistent with our newly added data that TTLL12 appears to be present at basal body and mitotic spindle centrosome, but not interphase, non-ciliated, cell centrosome. The idea that TTLL12 predominately regulates specialized microtubule structures (such as cilia and mitotic spindle) is also consistent with our data that TTLL12 strongly affects stability of mitotic spindle and cilia while having modest effect on interphase microtubule dynamics. We added few sentences to Discussion section to point that out.

12. The authors predominantly focus on the role of TTLL12 in regulating axonemal microtubules since it interacts with Rab19, which is required for proper ciliogenesis. However, when it comes to testing the functional mechanism of how TTLL12 regulates the whole process, they fail to show any evidence that is pertaining to the axonemes. Their studies on arginine methylation represented in Figures 6 and S4 are carried out either using porcine brain tubulin or the total pool of cellular tubulin. Whether methylated tubulin is present in the axonemes or not is something they have totally failed to address. The authors have to clearly demonstrate that the methylated tubulin is incorporated within the cilia, to emphasize that the effect of TTLL12 is majorly on axonemal microtubules.

We fully agree with the reviewer that so far we did not directly show that axonemal tubulin is methylated. That is hard experiment to do since we do not know exact methylation site on tubulin and consequently do not have anti-methyl-tubulin antibodies. Identification of methylation site and rising the antibodies are on our to do list, but we feel that it is beyond the scope of this manuscript. Thus, in this manuscript we use generic anti-methylation antibodies that recognize all methylated proteins, including histones that are very abundant in the cell. As the result, nuclear signal overwhelms any other cellular signal during microscopy analysis.

To partially address this reviewer concern we decided to immunoblot purified cilia. To achieve that we used a protocol developed by Wallace Marshal that uses calcium shock to de-ciliate epithelial cells (he used mIMCD-3 cells). The buffer containing released cilia is then subjected to series of gradient centrifugations. He used this prep to perform proteomic analysis of the cilia. We found that we can similarly de-ciliate MDCK cells (see new Supplemental Figure 5F), thus, we used the same protocol to purify cilia and blot it using anti-me1K, anti-me3K, anti1R, and anti3R antibodies. Importantly, we see the band at tubulin level with anti-me1K, anti1R, and anti3R but not anti-me3K antibodies (see Supplemental Figure 5F). We realize that this does not unequivocally prove that axonemal tubulin is methylated, but these data are consistent with proteomic analysis of porcine brain tubulin and suggest that ciliary tubulin may also be methylated.

Minor points:

1. Why do the authors think that TTLL12 is not a canonical Rab19 effector protein? There is no data in the manuscript that supports this claim. The authors could provide some clarity on their statement in the text or could show some data supporting their claim.

The main reason that we think that TTLL12 is not canonical Rab19 effector protein because its binding to Rab19 is not GTP dependent (see Supplemental Figure 1A, C, and D). Typically, Rab effector only bind to GTP-loaded (activated) Rab GTPases.

2. The IP-mass spec data shown in table S1 by the authors does not corroborate with their earlier IP done in 2021 (Jewett et al., 2021). If there is a direct interaction between TTLL12 and Rab19, then why do they not see a similar enrichment of Rab19 in their GFP-TTLL12 IP-mass spec?

- Is it due to the interference by GFP?
- Is it due to the differences in the experimental conditions used for the two IPs?
- The reason I ask this is because their data to show direct interaction is an in vitro system, while their IP is using lysates of MDCK cells expressing TTLL12. I wonder if the interaction is direct even in vivo / in cellulo, similar to what the authors see in vitro.

We do IP endogenous Rab19 with GFP-TTLL12 (see Figure 1B). Actually, both proteomic studies (earlier study in Jewett et al, 2021 and current study) are consistent with each other. Yes, the extent of precipitation is different, but that would be expected since in earlier study we expressed FLAG-Rab19 and IPed with anti-FLAG antibodies while in this study we expressed TTLL12-GFP and IPed with anti-GFP nanobodies.

3. What was the rationale behind choosing specifically TTLL12 from the different microtubule-associated proteins the authors identified in their Rab19 interactome? And what was the rationale behind selecting an enrichment of >6-fold enrichment? The authors need to clarify this in their manuscript.

The main rationale is that we identified endogenous TTLL12 in FLAG-Rab19 proteomics and endogenous Rab19 in TTLL12-GFP proteomics. Additionally, cilia microtubules are known to be heavily modified by other members of TTLL family. We edited result section to make that clear.

Choosing 6-fold enrichment is more arbitrary. In our proteomic studies we usually use cut-off enrichment value by calculation the enrichment of isolated protein (TTLL12 in this case) and dividing it by 2 (to make it a bit less stringent in order not to miss some binding partners). Since TTLL12 is enriched in GFP-TTLL12 sample ~13 fold (as compared to GFP-only, see Figure 1B), we chose cut-off value as 6. Importantly, in supplemental Table 1 we list all proteins IDed by proteomic analysis, regardless of enrichment. We added this explanation to method section.

4. Typographical error: Page 3: "...which exhibits high conservation as compared to mammalian tubulin, but has no tubulin tail PTMs...."

Error fixed.

5. Typographical error: Page 3: ".....TTLL12 and the microtubule polymer may not that be...." has to be: may be not that

Corrected.

6. The supplementary moves 1 and 2 provided by the authors does not clearly say which cell is the WT and which cell is the KO. IT is difficult to make out from the video to understand which one has the faster speed of polymerization as suggested by the authors in Figure 3I. The authors can slow down the speed of the video to be able to appreciate this aspect with better clarity.

As suggested we labeled both supplemental movies to make it clear which on is WT and which one is TTLL12 KO.

7. It was not clear to me why the authors analyzed the presence of transition zone in their TTLL12 KO cells, since the cells show normal cilia formation. This is already an indication that the transition zone is normal.

We feel that formation of the cilia does not necessarily mean that cilia is fully functioning. One way to test that (at least partially) is to see whether cilia forms transition zone. This is the reason why we stained for transition zone proteins. We would prefer to keep that data in the manuscript but can remove it if reviewer feels strongly about it.

8. The immunofluorescence microscopy image of Rpe1 cells stained with the TTLL12 polyclonal antibody in Figure 4G and Figure S3G are not very convincing. Suggesting that TTLL12 is localizing to the base of the cilia from this image is quite speculative as one of the cells with GT335 positive cilia in Figure 4G does not show any localization of TTLL12 at the base of the cilia. Moreover, from this image it is not conclusive whether the TTLL12 is at the base of the cilium or the tip, as centriolar markers are missing. It would be good to have a better representative image of this, with centriolar marker like gamma tubulin or centrin to highlight the base of the cilium.

We have also observed that not all cilia contain TTLL12 puncta. That is one of the reasons we decided to quantify the signal and compare to TTLL12 KO cells to ensure that TTLL12 localization to the basal body is real (see Figure 4H). As one can see from quantification, it is clear that fraction of the cilia has pretty low concentrations of TTLL12. We are not quite sure why is that the case. One intriguing possibility is that TTLL12 accumulation depends on the state of the cilia (growing or retracting), although further studies will be needed to fully understand that. Please note that in this revised manuscript we also included new data showing that TTLL12 is present at mitotic spindle centrosomes (Supplemental Figure 3A-B). These data further supports the idea that TTLL12 may function at the centrosome during ciliation or mitotic spindle positioning.

9. It is quite difficult to differentiate the signals for Ac-tub and actin in Figure 5A. The authors are recommended to use different colors like blue or white to denote one of these proteins for better visualization and understanding of the observations.

To make it easier to differentiate between ac-Tub, actin, and gamma-tub we now show higher magnification show insets in black-and-white for each individual protein.

10. Typographical error: Page 9: ".....depleting TTLL12 in cells does not increase affect tyrosination...." It should be does not affect.

Error fixed.

Referee #3:

The Manuscript from Ceglowski et al addresses the function of the TTLL12 protein in cilia formation and general MT PTMs and stability. TTLL12 was identified as a binding partner for Rab19 which the authors have previously shown is involved in cilia formation. TTLL12 is a member of a large TTLL family, some of which are known to localize to the basal bodies and affect cilia, yet TTLL12 remained poorly characterized and had no known enzymatic activity. The authors perform a number of biochemical analysis of MT binding, PTMs, and stability as well as imaging analysis of cilia formation in both RPE1 cells and polarized MDCK. They claim that TTLL12 has an important role in regulating MT stability and in regulating cilia formation in exposed cilia of polarized cells but not in the submerged cilia of RPE1 cells. There is a lot of nice data in this paper but I found the story to be a bit incomplete and confusing. While the authors admit and discuss the limitations of direct versus indirect effects of TTLL12 on tubulin PTMS, the end result is a confusing combination of phenotypes where the reader still has little understanding of the function of TTLL12. While the data is interesting, I think there needs to be a little more mechanistic insight before it is up to the level of EMBO Reports.

Comments:

1. One of the main claims of the paper is that TTLL12 has a functional difference on submerged versus polarized cilia. I think more needs to be done to substantiate this claim. It could be that these two cells have different levels of other TTLLs that compensate or some other cell type difference unrelated to the type of cilia. Mazo et al. showed that cilia submersion was regulated by sub distal appendages in RPE1 cells. Perhaps the best experiment I can think of to properly test the authors model would be to generate RPE1 cells that have an exposed cilia and to see if there is now a defect similar to MDCK. The formation of cilia in RPE1 cells that are not acetylated is intriguing. Are these cilia stable? Are they functional?

We do agree with the reviewer that it remains to be defined whether the presence of ciliary pocket is what determines the resistance of RPE1 ciliation to loss of TTLL12.

Unfortunately, as far as I know there is no good way to generate RPE1 cells that do not have ciliary pocket. Elimination of sub distal appendages would lead to all kind of other issues in addition to inhibition of cilia submersion. For example, sub distal appendages are presumably needed for cilia docking at plasma membrane in epithelial cells. Generally, it is believed that epithelial cells form their cilia in somewhat different way (as compared to non-polarized cells), known as extracellular pathway, that leads to them not having any ciliary pocket.

In this manuscript we do show that RPE1 TTLL12-KO cells take longer to make cilia, thus, presumably they are less stable, at least during cilia formation. Once formed, they appear to be normal since they form transition zone, although we did not look at their signaling capabilities.

2. The OE MDCK cell line is massively over expressed relative to the endogenous levels. Given that, 2 spectral counts for Rab19 is pretty unimpressive and might argue against a strong interaction. It would be nice to see what other proteins bind TTLL12, this data set seems pretty cherry picked and it would be nice to know what else it interacts with and perhaps with those counts the relevance of Rab19 spectral counts would be more clear.

We agree that 2 spectral count is not very impressive. However, we also did proteomic analysis of FLAG-Rab19 and found endogenous TTLL12 in much higher levels (41 and 45 spectral counts; Supplemental Figure 1A). Furthermore, to ensure that Rab19 and TTLL12 binding is real, we have done in vitro binding assays as well as immunoprecipitation assays (see Supplemental Figure 1C-D, and Figure 2G-H).

To this manuscript we also added Supplemental Table 1 that lists all proteins identified in GFP-TTLL12 proteomics.

3. 2b...given the variability in this assay result it seems worthwhile to perform several more iterations. It seems like the one outlier might be affecting an important interpretation.

We have done this assay quite a few times since we originally hypothesized that TTLL12 binds to microtubules (various re-iterations of this assay were not included in the manuscript). We did, however, include the same assay using purified yeast tubulin to ensure that porcine brain tubulin post-translational modifications do not interfere with TTLL12 binding (see Supplemental Figure 1G-H). Finally, we did in vitro pull-down and immunoprecipitation assays at 4C to ensure that microtubules are depolymerized (Supplemental Figure 1F, Figure 2C and D). All of these data clearly suggest that TTLL12 binds to tubulin heterodimer rather than microtubules. That is also consistent with our imaging data where we never see TTLL12 localize to microtubules.

4. Are dimers actually acetylated? I feel like there is a real conceptual break in what the authors are proposing. It is simply not clear how a protein that only interacts with non polymer MTs is affecting MT polymers. While the authors discuss this at length without

experimental insight we simply don't know what TTLL12 is doing. They have observed a lot of changes to MTs but what is direct and what is indirect are left unanswered.

No tubulin heterodimers are not acetylated. This is due to the fact that tubulin de-acetylases, such as HDAC6, bind preferentially to heterodimers. In contrast, tubulin acetylases, such as TAT binds preferentially to microtubules.

Regarding how enzyme that binds to heterodimer can affect microtubules, we propose that TTLL12 methylates heterodimers that are then incorporated into microtubule lattice and then affect microtubule stability (see our Discussion section). Presence of TTLL12 at the basal body and mitotic spindle centrosomes would place TTLL12 in perfect location to do just that during formation of cilia and/or mitotic spindle. Indeed, the most dramatic effects of TTLL12 KO are observed on spindle positioning and cilia extension rather than other interphase microtubules. We edited our Discussion section to make that clear. We also realize that we did not unequivocally prove that TTLL12 is methyltransferase, although several pieces of data strongly suggests that. This is why we tend to be cautious during our final conclusions and proposed model. However, we do feel that this manuscript lays a good foundation (and reports interesting findings) for future studies of TTLL12 involvement in microtubule dynamics, ciliation, and mitotic spindle formation by our and other laboratories.

5. In MDCK cells the rescue causes a dramatic increase in ciliation, which is likely more of an overexpression than a rescue but is an interesting result that should be discussed.

As suggested, we added discussion about this result.

6. "We propose that TTLL12 actively functions at the basal body and interacts with the α/β -tubulin heterodimer before incorporation into the primary cilium; this interaction promote microtubule stability and tubulin PTMs required for building an axoneme (Fig. 7)"....Perhaps the most fascinating data presented is the fact that the RPE cells generate a cilium without acetylation, which is pretty amazing, but is also inconsistent with this statement. Additionally, the fact that Arl13b is at the centrosome in MDCKs, suggests that cells are starting to make cilia but that they are just short and underdeveloped rather than unciliated.

We do agree with the reviewer that cell may still form short and underdeveloped cilia in TTLL12 KOs. Thus, we edited our "Discussion" section to accommodate this possibility. Thus, aforementioned statement now states, "We propose that TTLL12 actively functions at the basal body and interacts with the α/β -tubulin heterodimer before incorporation into the primary cilium; this interaction promote microtubule stability and regulate (likely indirectly) tubulin acetylation required for extension and maintenance of the axoneme (Fig. 8)".

We also agree that inhibition of ciliation in TTLL12 KOs is one of more intriguing findings in this manuscript. It is also one of more puzzling findings since we still do not quite understand how acetylation is affected (presumably indirectly). Thus, while we

discuss these findings in “Discussion” section, I am reluctant to include that in our proposed model. We are continuing to work on it and perhaps in future papers we can propose a better model that does incorporate tubulin acetylation.

Dear Prof. Prekeris,

Thank you for submitting your revised manuscript. It has now been seen by two of the original referees.

My colleague Ioannis moved on to The EMBO Journal, therefore I stepped in as the handling editor of your manuscript.

The referees find that the study is significantly improved during revision and recommend publication. However, I need you to address the points below before I can accept the manuscript. Given the nature of the revisions, we believe the manuscript can be submitted by the 31st of October. Please let me know if you foresee a problem in meeting this deadline.

- Please discuss the results concerning the lack of TTL12 KO rescue by TAT1 as pointed out by referee #2 (paragraph 3).
- Please address the remaining concern of referee #3 on Figure 3A either textually or by synchronizing the cells.
- Please address the remaining concern of referee #3 on Figure 6E by providing images with higher contrast.
- Please provide 3-5 keywords for your study. These will be visible in the html version of the paper and on PubMed and will help increase the discoverability of your work.
- As per our guidelines, please add a 'Data Availability Section', where you state that no data were deposited in a public database.
- Please rename the 'Conflict of interest' as 'Disclosure and Competing Interests Statement'.
- Please rename "Methods and Protocols " as "Materials and Methods".
- If possible, the names in the manuscript should be displayed as they are in the manuscript submission system. Currently all the names in the ms are listed as last name, comma, first name initial: e.g. Ceglowski, J in ms file vs. Julia Ceglowski in the the manuscript submission system.
- We note that the funding information is not complete in the manuscript submission system (Bolie Scholar Award is missing).
- Source data need to be resubmitted as one file per figure (source data of one figure to be grouped into one folder and then each zipped folder should be separately uploaded). As per figures 3B, D, 4A-B, G, 5A, B D, F, I, please do submit all images used for quantification.
- Please remove source data from the Supplemental file / Appendix.
- We note that currently there are 8 Supplemental figures separately uploaded. Up to 5 can be converted to EV figures, in which case they need to be renamed to EV Figures 1-5. The remaining figures would then need to be included into a single Appendix file with a Table of Contents and page numbers. The appendix figures would need to be called Appendix Figure S1, etc. and callouts in the manuscript also need to be updated accordingly. Suppl. figure legends need to be removed from the ms and provided in the Appendix file, unless there will be some EV figures in which case, we keep the legends in the manuscript.
- Supplemental Table 1 and 2 should be renamed to Dataset EV1 and EV2 and should be uploaded as datasets; the callouts in the ms need to be updated accordingly.
- The legends of the movies need to be removed from the manuscript; each legend should be zipped up with its corresponding movie in one folder and uploaded. The legend can be in readme.txt file. The correct nomenclature and callouts are Movie EV1, etc.
- Our data editors have asked you to clarify the below points in the figure legends:
 - o Please indicate the statistical test used for data analysis in the legends of figures 2b, d, f, h; 3a, c, e, i; 4c, d, f, h; 5c, e, g, h; 6b-c; 7c, e, g; supplementary figures 1d, h; 2a-b, d-e, g; 3c-e; 4f, h.
 - o Please note that information related to n is missing in the legend of figures 2b, d, f, h; 3i; 4c, d, f, h; 6b-c, f; 7c; supplementary figures 1d; 2g
 - o Please describe the nature of entity for 'n' in the legends of figures 7b; supplementary figures 2a-b, d-e; 3c; 4e; 5a-c.
 - o Please note that the error bars are not defined in the legend of figures 7b-c; supplementary figures 2g; 4e-f; 5a-c.
 - o Please indicate what white arrows represent in supplementary figure 3a-b; 5f.
 - o Please note that scale bar and its definition are missing for supplementary figure 4d.
- Papers published in EMBO Reports include a 'synopsis' and 'bullet points' to further enhance discoverability. Both are displayed on the html version of the paper and are freely accessible to all readers. The synopsis includes a short standfirst summarizing the study in 1 or 2 sentences (max 35 words) that summarize the paper and are provided by the authors and streamlined by the handling editor. I would therefore ask you to include your synopsis blurb and 3-5 bullet points listing the key experimental findings.
- In addition, please provide an image for the synopsis. This image should provide a rapid overview of the question addressed in the study but still needs to be kept fairly modest since the image size cannot exceed 550 (width) x 300-600 (height) pixels.

Thank you again for giving us to consider your manuscript for EMBO Reports, I look forward to your minor revision.

Kind regards,

Deniz Senyilmaz Tiebe

--

Deniz Senyilmaz Tiebe, PhD

Referee #2:

In their revised manuscript, Ceglowski et al have provided additional evidence to suggest that TLL12 is an interactor of Rab19, the small GTPase which is a critical factor of ciliogenesis. The authors also provide new evidence to suggest a role for TLL12 in providing stability to the microtubules through its enzymatic activity wherein MDCK cells KO for TLL12 are more susceptible to nocodazole treatment and this can be rescued by expressing TLL12 in the KO background. This is a very interesting observation that can have significant cellular implications. Moreover, the authors also provide evidence to suggest that TLL12 is present at the centrosomes and that the enzyme activity is essential for spindle stability.

One of the highly interesting observations in this manuscript is that while MDCK cells KO for TLL12 lose their ability form longer cilia, the same is not observed in case of RPE-1 cells KO for TLL12. This suggests that cells having a ciliary pocket can overcome the loss of TLL12 for their ciliogenesis, while those having an extracellular ciliogenesis pathway may require TLL12 to stabilize the axonemes for proper ciliogenesis. This again provides a new mechanistic insight into the impact of the tubulin code in regulating specific microtubules.

The authors also provide new evidence in their revised manuscript that suggests that TLL12 localizes to the migrating end of the cells and thus regulates cell migration. The observation that loss of TLL12 in RPE-1 cells leads to faster cell migration compared to the wildtype cells is consistent with earlier studies that have implicated tubulin acetylation in regulating cell migration. However, it is worth noting that in these cells, it appears that the loss of acetylation due to TLL12 KO is not rescued by α TAT1, the canonical tubulin acetylase. Whether the two enzymes play different roles in terms of this migratory process, or whether it is a cell-dependent process, is an interesting aspect the authors have not investigated here, or commented upon.

Overall, the manuscript provides a novel, fundamental insight into the role of TLL12 with an implication for its role in axoneme biogenesis, dynamics, stability and in regulating specific tubulin PTMs that are divergent from those that other enzymes of the TLL family catalyse. Moreover, the authors provide additional interesting insights into the role of TLL12 in spindle stability as well as in cell migration. Whether this is regulated by its role in regulating acetylation or methylation or both is an interesting proposition the authors will need to investigate in the future. The manuscript is well-written and the data is very succinctly outlined.

They have also addressed all the concerns I had in the earlier version of the manuscript and I have no further comments to add.

Referee #3:

The authors have attempted to address many of the reviewers concerns and there has been an incremental improvement over the previous submission. The revised manuscript however, failed to improve on the lack of mechanistic insight into the function of TLL12. As I mentioned in the original review there is a lot of high quality data, but the lack of mechanistic advance lessens my excitement for this paper.

Comments:

At the request of reviewer 2 the authors have replaced Figure 3A. While the original figure was not particularly convincing, this new one is even less so. If the authors feel this is a critical point worth making they should synchronize the cells as to focus on the spindle MTs of mitotic cells as they do in 3B-E. Other than that it is a weak result that should probably be de-emphasized as their model argues that whole cell analysis of tubulin acetylation is not relevant.

The contrast on 6E is not sufficient to see the cells.

The addition of the new protocol to isolate cilia for western analysis is really a missed opportunity to test aspects of their model. Overexpression or KO of TLL12 could alter the methylation and this could be assayed using this approach.

-Please discuss the results concerning the lack of TTL12 KO rescue by α TAT1 as pointed out by referee #2 (paragraph 3).

Discussion added (see marked in yellow).

-Please address the remaining concern of referee #3 on Figure 3A either textually or by synchronizing the cells.

Since it is technically difficult to synchronize MDCK cells, we added discussion regarding Figure 3A (marked in yellow). Please note that decrease in tubulin acetylation in MDCK cells is quite moderate (see quantification). Consequently, decrease in shown western blot is also quite moderate (but clearly visible).

Please address the remaining concern of referee #3 on Figure 6E by providing images with higher contrast.

Images with higher contrast added.

-Please provide 3-5 keywords for your study. These will be visible in the html version of the paper and on PubMed and will help increase the discoverability of your work.

Keywords added.

-As per our guidelines, please add a 'Data Availability Section', where you state that no data were deposited in a public database.

Added.

-Please rename the 'Conflict of interest' as 'Disclosure and Competing Interests Statement'.

Changed.

-Please rename "Methods and Protocols " as "Materials and Methods".

Changed.

-If possible, the names in the manuscript should be displayed as they are in the manuscript submission system. Currently all the names in the ms are listed as

last name, comma, first name initial: e.g. Ceglowski, J in ms file vs. Julia Ceglowski in the the manuscript submission system.

Changed.

-We note that the funding information is not complete in the manuscript submission system (Bolie Scholar Award is missing).

Bolie Scholar Award added to manuscript submission system.

-Source data need to be resubmitted as one file per figure (source data of one figure to be grouped into one folder and then each zipped folder should be separately uploaded). As per figures 3B, D, 4A-B, G, 5A, B, D, F, I, please do submit all images used for quantification.

Images have been added to figure source files. Please note that these are raw image files from Nikon Software. Also, please note that images shown in Figure 5I were not used for quantifications and are the only images available in the manuscript.

-Please remove source data from the Supplemental file / Appendix.

Removed.

-We note that currently there are 8 Supplemental figures separately uploaded. Up to 5 can be converted to EV figures, in which case they need to be renamed to EV Figures 1-5. The remaining figures would then need to be included into a single Appendix file with a Table of Contents and page numbers. The appendix figures would need to be called Appendix Figure S1, etc. and callouts in the manuscript also need to be updated accordingly. Suppl. figure legends need to be removed from the ms and provided in the Appendix file, unless there will be some EV figures in which case, we keep the legends in the manuscript.

Five supplemental figures were converted to EV figures. Rest were added as source data to corresponding Figure Source files.

-Supplemental Table 1 and 2 should be renamed to Dataset EV1 and EV2 and should be uploaded as datasets; the callouts in the ms need to be updated accordingly.

Renamed.

-The legends of the movies need to be removed from the manuscript; each legend should be zipped up with its corresponding movie in one folder and uploaded. The legend can be in readme.txt file. The correct nomenclature and callouts are Movie EV1, etc.

Done as instructed.

-Our data editors have asked you to clarify the below points in the figure legends:
a) Please indicate the statistical test used for data analysis in the legends of figures 2b, d, f, h; 3a, c, e, i; 4c, d, f, h; 5c, e, g, h; 6b-c; 7c, e, g; supplementary figures 1d, h; 2a-b, d-e, g; 3c-e; 4f, h.

Information about statistical test added to the figure legends.

b) Please note that information related to n is missing in the legend of figures 2b, d, f, h; 3i; 4c, d, f, h; 6b-c, f; 7c; supplementary figures 1d; 2g

Information related to n was added.

c) Please describe the nature of entity for 'n' in the legends of figures 7b; supplementary figures 2a-b, d-e; 3c; 4e; 5a-c.

Explanation added.

d) Please note that the error bars are not defined in the legend of figures 7b-c; supplementary figures 2g; 4e-f; 5a-c.

Requested information added to figure legends.

e) Please indicate what white arrows represent in supplementary figure 3a-b; 5f.

Added.

f) Please note that scale bar and its definition are missing for supplementary figure 4d.

Scale bar added.

-Papers published in EMBO Reports include a 'synopsis' and 'bullet points' to further enhance discoverability. Both are displayed on the html version of the

paper and are freely accessible to all readers. The synopsis includes a short standfirst summarizing the study in 1 or 2 sentences (max 35 words) that summarize the paper and are provided by the authors and streamlined by the handling editor. I would therefore ask you to include your synopsis blurb and 3-5 bullet points listing the key experimental findings.

Added to title page of the manuscript.

-In addition, please provide an image for the synopsis. This image should provide a rapid overview of the question addressed in the study but still needs to be kept fairly modest since the image size cannot exceed 550 (width) x 300-600 (height) pixels.

Synopsis image uploaded.

Dear Prof. Prekeris,

Thank you for submitting your revised manuscript. I have now looked at everything and all is fine. Therefore, I am very pleased to accept your manuscript for publication in EMBO Reports.

Congratulations on a nice work!

Kind regards,

Deniz Senyilmaz Tiebe

--

Deniz Senyilmaz Tiebe, PhD

Editor

EMBO Reports

--
